# Compact Modeling Framework v3.0 for high-resolution global ocean-ice-atmosphere models

Vladimir V. Kalmykov[1,3], Rashit A. Ibrayev[1,2,3,4,5], Maxim N. Kaurkin[1,3,5], and Konstantin V. Ushakov[1,2,3,5]

[1]Hydrometcenter of Russia, B. Predtechensky per., 11-13, Moscow, 123242, Russia
[2]Marchuk Institute of Numerical Mathematics, Russian Academy of Sciences, ul. Gubkina, 8, Moscow, 119333, Russia
[3]Shirshov Institute of Oceanology, Russian Academy of Sciences, Nahimovskiy prospekt, 36, Moscow, 117997, Russia
[4]Moscow Institute of Physics and Technology (State University), Institutskiy per. 9, Dolgoprudny, Moscow oblast, 141700, Russia
[5]Federal State Budget Scientific Institution "Marine Hydrophysical Institute of RAS", Kapitanskaya str., 2, Sevastopol, 299011, Russia

**Correspondence:** Maxim N. Kaurkin (maksim.kaurkin@phystech.edu)

**Abstract.** We present a new version of the Compact Modeling Framework (CMF3.0) developed for the software environment of stand-alone and coupled global geophysical fluid models. The CMF3.0 is designed for use on high and ultra-high resolution models on massively-parallel supercomputers.

The key features of the previous CMF version (2.0) are mentioned to reflect progress in our research. In the CMF3.0, the MPI approach with a high-level abstract driver, optimized coupler interpolation, and I/O algorithms is replaced with the PGAS paradigm communications scheme, while the central hub architecture evolves to a set of simultaneously working services. Performance tests for both versions are carried out. As an addition, some information about the parallel realization of the EnOI (Ensemble Optimal Interpolation) data assimilation method and the nesting technology, as program services of the CMF3.0, is presented.

## 1 Introduction

As it was stated at the World Modeling Summit for Climate Prediction (Shukla, 2008), there is a general agreement that much higher resolution of the major model components (atmosphere, ocean, ice, land) is a fundamental prerequisite for a more realistic representation of the climate system and more relevant predictions (e.g., extreme events, convection, tropical variability, etc.).

Along with the development of physical models of individual Earth system components, the role of instruments organizing their coordinated work (couplers and coupling frameworks) becomes more and more important. The coupler architecture depends on complexity of the models used, on the characteristics of interconnections between the models and on the hardware and software environment. Historically, the development of couplers follows the development of coupled atmosphere-ocean models. At some level of complexity, the development of such software became an external problem relative to the development of individual components of the coupled model.

The first coupled models used simple algorithms for coordination of components through the file system. There was no separate coupler component, and communication between models was realized as a set of model procedures for input/output (I/O) and for interpolation between global model grids (today this method is used, for example, in INMCM4.0 climate model (Volodin et al., 2010)). At the next stage, the coupling of components was done through a separated central sequential hub using the multiple executable approach (OASIS3 (Valcke, 2013), Community Climate System Model cpl3 (Craig et al., 2005)).

Coupling through a shared file or through a sequential hub is acceptable only for models of relatively low resolution. Increasing of array sizes and of the number of model components in the system will inevitably become a "bottleneck" because of memory and performance limitations of a single processor core and also due to problems related to global network communications. Therefore, it was quite natural that the next generation of couplers introduced parallelism in their internal algorithms (Community Earth System Model cpl6 (Craig et al., 2005), OASIS4 (Redler et al., 2010)). The parallel coupler architecture solves computational problems for fine grids, but increases complexity of algorithms.

A new coupler architecture was introduced for the CESM1.0 model in 2012 (Craig et al., 2012). In this system, the coupled model has the form of a single executable and contains a high-level driver that calls a few standard component subroutine interfaces (init, run, finalize, etc.). This approach requires some reorganization of the components' code and its adaptation to the interfaces understandable by the driver, but simplifies model synchronization.

The coupled system also can be launched as a single or multiple executable without a separate coupler, whose functions in this case are provided by a coupling library and performed in parallel on a core subset of each model component. Such a solution was proposed in OASIS3-MCT (Craig et al., 2017). A high-level driver controlling system sequencing is not required in this case.

Another important feature of the coupled model is the scheme of working with the file system. In earlier versions, it was carried out independently by each model component in a sequential way. Obviously, this master-process scheme (used in CESM cpl6, OASIS3, OASIS3-MCT) was limited by the RAM of a node. Increasing amounts of model data lead to rapid development of parallel I/O algorithms. Since version 1.0, the CESM system utilizes the PIO library (Dennis et al., 2012a) to establish parallel output in the NetCDF format by every component through several cores that play the role of writing delegates. In the GFDL FMS system (Balaji, 2012), fully parallel data storage with file post-processing at the end of the run is offered.

Thus, we can point out the necessary features of modern coupling frameworks, which define their functionalities and characteristics:

1. coupling architecture (serial, parallel, with a high-level driver or as a set of procedures); the design of the framework defines the complexity of development/maintenance of the coupled model and implicitly establishes performance limitations;

2. I/O-module architecture (serial or parallel, synchronous or asynchronous); it should be considered as a balance between simplicity of algorithms and the necessary rate of I/O;

3. ease of use; the level of system abstraction defines the convenience of user's work and the transparency of the overall coupled model;

4. performance; the choice of underlying algorithms defines the computational rate of the coupled model.

## 2 Background

Our work began with the development of a parallel version of an ocean dynamics model. The aim at that time was to work out a high-resolution World Ocean model (WOM). We had to solve several problems, namely halo update, mapping (interpolation) of external forcing data to the model grid, saving solution to a file, and gathering diagnostics. It was obvious that separation of numerical algorithms for solving ocean dynamics equations from low-level service procedures is necessary to write a transparent code, which would allow us to develop independently the physical model as well as service procedures.

This approach showed its advantages in coupling the atmosphere and ocean general circulation models for medium- and long-term weather forecasts at the Hydrometeorological Research Center of Russia. The purpose was to create the software capable of maintaining effective interaction of the high-resolution (on the order of 0.1 degrees) models of atmosphere and ocean with a possibility to extend the coupled model by incorporating ice and soil components. The components of the coupled model were the INMIO World Ocean model (Ibrayev et al., 2012) based on the MESH sea hydrodynamics code (Ibrayev, 2001) and the SLAV Global atmosphere model (Tolstykh et al., 2017). It turned out that for coupling of several models one should solve similar problems as for a standalone model (mapping, I/O), but also has to provide synchronization and consistency of the interpolated data for simultaneously running components.

At the beginning of our study in 2012 there were several solutions for creation of coupled models. It should be noted that state-of-the-art couplers, such as of CESM (with coupler based on MCT (Larson et al., 2005) or ESMF (Theurich et al., 2016) packages) and OASIS, are fairly complex programs. The CESM cpl 7 (Craig et al., 2012) is written for a predefined set of components, and introducing a new model requires non-trivial changes and some work with internal structures. Adding a new grid still requires non-automated constructing of interpolation weights for it (CESM). Tests showed that computational costs of the CESM coupler (including coupling, remapping and surface flux computation) are quite significant 20% (Craig et al., 2012); nevertheless, good results of 2.6 SYPD (Simulated Years Per wall-clock Day) rate were achieved for the ultra-high resolution Earth model (Dennis et al., 2012b).

The OASIS3 system was very successful and was widely used by many research groups around the world. But, as it was pointed out, it contains a serial coupler, which is an obvious performance bottleneck due to constraints on memory and global communications. The new version, OASIS3-MCT (Craig et al., 2017) resolves the issue of sequential interpolation by using MCT procedures executed on all model component cores, instead of mapping through a standalone coupler. Unquestionable advantage of this no-standalone-coupler design is the minimization of interference in the user code, since there is no need to adapt it to interfaces required by a coupler. But still the system contains master-process I/O routines, which can be a limiting factor in case of intensive data dumping. Even with parallel I/O, the solution with a subset of service processes provides double load on the model cores, which at the same time perform solving of the model equations, coupling actions and I/O-tasks. Nevertheless, such behaviour does not limit the coupling and mapping on large grids.

According to the analysis of the Coupling technologies for Earth System Modeling workshop (Valcke et al., 2012), today there are several common aspects in coupling software development: an ability to communicate data between components, regrid data, and manage the time evolution of the model integration. There is a lot of custom parallel coupling mechanisms, with either single or multiple executable approaches. We selected the approach with a single executable because it can simplify the program flow and give additional opportunities for performance optimization. Besides, we used the NetCDF standard for parallel I/O and SCRIP (Spherical Coordinate Remapping and Interpolation Package) (Jones, 1999) for regridding, as done in OASIS3 (Valcke, 2013).

In the CMF2.0, the framework for ocean-ice-atmosphere-land coupled modeling on massively-parallel architectures (Kalmykov and Ibrayev, 2013), we implemented these basic ideas.

In this paper we present two versions of the Compact Modeling Framework (CMF), v. 2.0 and v. 3.0. As the CMF2.0 was published only in Russian (Kalmykov and Ibrayev, 2013), here we outline the basics of that version. In the CMF2.0 we combine the common proposals of the Earth system modeling community and an experimentation with low-level algorithms. We focus on the single executable hub approach with a high-level abstract driver, optimized interpolation algorithms, asynchronous I/O routines, and tools for pre- and post-processing stages.

In the CMF3.0, the pure MPI approach is replaced with the Partitioned Global Address Space (PGAS) paradigm of communications, while the central hub architecture has evolved to a kind of service-oriented architecture (SOA) with a set of simultaneously working services and a common task queue.

## 3 CMF2.0 overview

### 3.1 Architecture of the coupled system

Any coupled model under control of the CMF runs as a single executable, with each model component and the coupler using distinct processor cores. At the beginning, the global MPI-communicator is split into appropriate groups according to the requested communicator sizes of the model components and the coupler, and then all groups work simultaneously. The coupler performs some initialization routines and enters the time cycle of requests. Following the *Template method* (Gamma et al., 1995), all model components do the same logical steps by calling predefined abstract interfaces, for example, *ini_grid, ini_data, main_step, finalize*. Meanwhile, the CMF system does not know, what particularly will be done inside these routines. The realizations of abstract interfaces represent the specific behaviour of every model component: initializations and registration in the system of all data arrays that will be involved in component-component exchanges and in I/O; the main step of physics equations solving for the particular model component; finalizing procedures, etc.

That is, in order to add a physical model to the coupled system a user only has to define the physical model adapter (the required template is provided) and to realize its abstract interfaces (filling them with calls to his internal model subroutines). This approach allows one to generate different executables for different coupled model combinations (e.g., switch between ocean simulations with different sea ice models) and restricts the user from any changes in the code outside of his adapter. Also

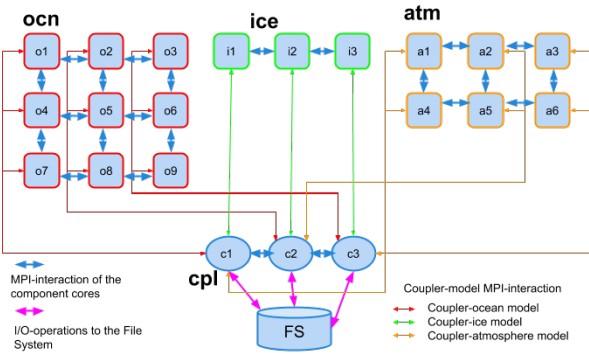

**Figure 1.** Architecture of the coupled model run under control of the CMF2.0. In this example there are three components (ocean, atmosphere, ice) connected by the 3-core coupler.

the addition or modification of components does not affect the main CMF code, because it implements the abstract driver and the abstract component, which do not mention any specific component names (ocean, atmosphere, ice, etc).

### 3.2 Coupler-model interactions

For any model component, its decomposition is generated by the CMF2.0 system in such a way that each coupler core interacts only with a specific subset of the component cores. This allows to reduce the required amount of communication routes to the coupler for every component from $M \times N$ (which would be in the general case of the component and the coupler running on $M$ and $N$ cores, respectively) to only $M$, since each component core ("slave") now interacts just with its "master" coupler core. Therefore, the size of every component communicator has to be a multiple of the coupler communicator size. In order to

meet this condition in practice (e.g., for "poorly divisible" component grid sizes), the CMF2.0 partially supports uneven grid subdomain sizes by making the last row of the 1D- or 2D-decomposition narrower than the others. An example of the coupled model with 3 coupler cores and 3 components is shown on Fig. 1.

All events in the system are divided into few classes (save diagnostics, save control point, read file data, send/receive mapping, etc.), defining different actions with data arrays. In the CMF2.0, we postulate that all events could be predefined

before the start and occur with fixed periods. Thus, the coupler can take on the task of synchronizing models and avoiding deadlocks.

The sequence of events (time chain) is constructed in the main CMF program, which is the entry point of the coupled model. Also, at the registration stage, models provide the CMF system with pointers to the arrays that must be processed in the events. So, during the system operation, events are performed automatically and do not require explicit calls from the user. As the

information about the periods of all events is known at the registration stage, the coupler can build a table of its actions. This

allows to exclude parallel synchronization of the coupler cores, which otherwise would be necessary when, for example, two components at the same time want to write data to the file system. When a certain time moment arrives, the coupler selects the next event from the chain and calls the appropriate handler function based on the type of this event, while the model components asynchronously send data. Moreover, it becomes possible to use persistent MPI-operations (combinations of *MPI_SEND_INIT*
and *MPI_STARTALL*) for all events, thus saving time of repeated communications. Combination of predefined time chain, persistent communications and pointer-based asynchronous sending provides high efficiency of parallel data gathering and distribution.

## 3.3 Coupler: mapping

The interpolation algorithm uses SCRIP-formatted weight files built at the pre-run (off-line) stage by means of the CDO package (http://mpimet.mpg.de/cdo). At the beginning of the run stage, the weight files are read by the coupler in parallel. During the run, the data intended for mapping is sent by each component core to its master core asynchronously, without blocking.

The regridding process is performed in the coupler communicator and is implemented as a sparse-matrix – vector multiplication. It supports logically-rectangular grids. We implemented the "source" and "destination" parallel mapping algorithms (Craig et al., 2005), which correspond to the weights being distributed according to the destination or source grid decomposition, respectively. The former is usually more efficient if the source grid has fewer points than the destination one, and the latter vice versa.

The SCRIP format is used to organize the mapping process, i.e. SCRIP-type links connect cells of destination and source grids with appropriate weights. Since every coupler core works only with a subdomain of the global model grid, it has only a part of the source grid data in memory. Other data should be gathered from neighbour cores during every interpolation event, which is functionally analogous to calling the Rearranger routines of (Jacob et al., 2005). Both the component-component SCRIP links and intracoupler rearrange routes are initialized at the beginning of the run and used at the run stage as persistent.

During the interpolation event, every coupler core first prepares and sends source cells required by its neighbours. Then, while this data is being sent, it weights its local source cells. And, at last, receives the missing data and completes the weighted sums on the destination grid. It is worth noting, that the data is not sent directly, but as sorted unique cell vectors. This allows one to avoid sending duplicated data which could be the case when a source cell is used in a few destination cells. As a result, there is an overlap of computations and communications, which, in conjunction with persistent MPI transactions, determines high efficiency of the algorithm.

The performance rate of the CMF2.0 interpolation system was evaluated in several "ping-pong" tests, in which the coupler was ensuring component-component exchanges of the INMIO-SLAV ocean-atmosphere model with disabled solvers of physics equations (similarly to the ping-pong test of OASIS3 in (Valcke, 2013)).

In the Test I, the ocean model sends three 2D-fields every 2 hours to the atmosphere model and receives nine 2D-fields every 1 hour. The ocean model has the $3600 \times 1728$ tripolar grid and the atmosphere model has the $1600 \times 864$ latitude-longitude grid (grids were taken from the current versions of ocean (Ibrayev et al., 2012) and atmosphere (Tolstykh et al., 2017) models).

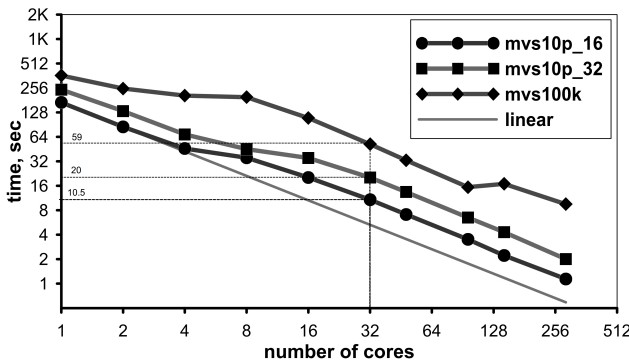

**Figure 2.** Walltime required for the 10-day ocean-atmosphere model run with disabled physics vs. number of coupler cores on MVS super-computers (Test I for CMF2.0).

The mapping process consists of gathering data from the source component, regridding inside the coupler communicator and distributing the result to the destination component. The test was run for 10 model days, which corresponds to $120 \times 3$ ocean-atmosphere mappings and $240 \times 9$ atmosphere-ocean mappings. Sizes of communicators for the ocean and atmosphere model components were fixed by 1152 and 288 cores, respectively. While not performing any computational work, they allow to imitate real communication load of the overall system, reflecting packing, MPI sending, and unpacking costs. Thus, the charts present a strong scalability of the coupler interpolation algorithm.

Results were obtained on four supercomputers: MVS-100k, MVS-10P, BlueGene/P, BlueGene/Q (characteristics are provided in the Appendix). On all supercomputers, the coupled system was compiled with a standard Intel Fortran compiler. Timing results of the 10-day Test I on MVS supercomputers are presented in Fig. 2. Two configurations, with 16 real and 32 virtual cores per node, are shown for the MVS-10P. The difference in the speed of their work is expected and is a result of increased communication load for a larger number of cores per node. The graph shows good scalability with increasing size of the coupler communicator. The best result of 1 second is achieved at 288 coupler cores.

It is clear that 20-40 coupler cores provide a satisfactory speed for such problems, because ~10 seconds costs for 10 model days is a rather insignificant value for the high-resolution ocean-atmosphere coupled modeling. The figure also shows inef-fectiveness of the sequential algorithm: even on the fast MVS-10P processors the service activity takes about 200 seconds. Besides, the work of sequential algorithm is only possible if all the node memory is allocated for the interpolation block, which is unlikely in practice (in our test we had to switch off physical model arrays allocation). Good coupler performance for one component-component connection is necessary for overall performance with growing number of components and their grid resolution.

Results of the same test for the BlueGene supercomputers are presented in Fig. 3. Timing of the algorithm is weaker than on the MVS-10P because of lower individual processor rate.

The Test II was conducted for estimation of the increasing communication load associated with the growth of components' communicator sizes. The timing still refers to the 10-day experiment with disabled physics. But the model grids were decom-

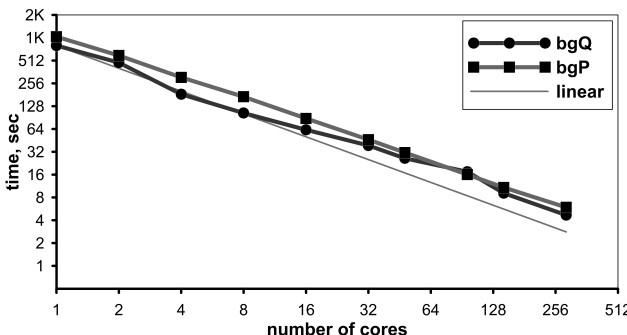

**Figure 3.** Walltime required for the 10-day ocean-atmosphere model run with disabled physics vs. number of coupler cores on BlueGene supercomputers (Test I for CMF2.0).

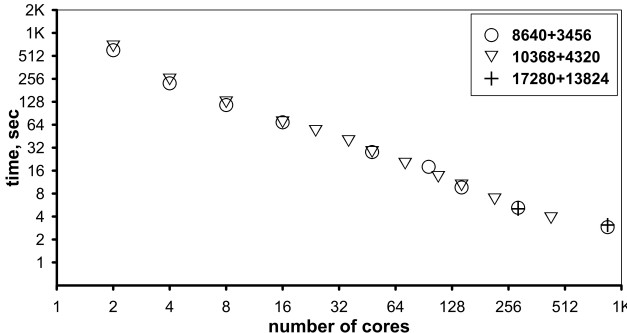

**Figure 4.** Walltime required for the 10-day ocean-atmosphere model run with disabled physics vs. number of coupler cores on BlueGene/Q supercomputer, for different decomposition sizes of the ocean and atmosphere models (Test II for CMF2.0).

posed on much higher number of subdomains, increasing the cost of gather/distribute phase of the test (mapping process inside the coupler communicator remains the same). The results are shown in Fig. 4 (curves replaced by point symbols to improve the readability of the graph) and, for convenience, repeated in the Table 1. Numbers of cores used for the ocean and atmosphere models were equal to 8640 and 3456, 10368 and 4320, 17280 and 13824, respectively.

5    The graph shows two interesting facts. Firstly, single-core coupler configurations do not work for the Test II because of memory limitations. Secondly, increasing of communication load (i.e. gather/distribute) affects performance only on small numbers of coupler cores. For example, test times for two coupler cores for model communicator sizes (8640, 3456) and (10368, 4320) are correspondingly 26% and 42% higher than for Test I communicator sizes (1152, 288). For eight coupler cores this difference becomes 12% and 21%, correspondingly. Since every coupler core communicates only with a subset of component cores, increasing of the coupler communicator size leads both to decomposing of the interpolation computations and to decreasing of the component-coupler communication overhead, though slightly increasing intracoupler rearrangement communications. As a result, even few tens of coupler cores are suitable to provide good performance of high-resolution mapping with huge sizes of model communicators.

**Table 1.** The same data as in Fig. 4, with added BlueGene/Q data from Fig. 3.

| OCN+ATM cores → | 1152+288 | 8640+3456 | 10368+4320 | 17280+13824 |
|:---:|:---:|:---:|:---:|:---:|
| CPL cores ↓ | | Time, sec | | |
| 2 | 481 | 608 | 681 | - |
| 4 | 184 | 224 | 250 | - |
| 8 | 104 | 116 | 126 | - |
| 16 | 62 | 69 | 69 | - |
| 24 | - | - | 53 | - |
| 36 | - | - | 39 | - |
| 48 | 26 | 28 | 28 | - |
| 72 | - | - | 19.7 | - |
| 96 | 17.5 | 18.0 | - | - |
| 108 | - | - | 13.2 | - |
| 144 | 9.1 | 9.6 | 10.2 | - |
| 216 | - | - | 6.7 | - |
| 288 | 4.7 | 5.2 | - | 5.0 |
| 432 | - | - | 3.8 | - |
| 864 | - | 2.9 | - | 3.1 |

"-" denotes not tested or unsupported configurations

## 3.4 Input/output scheme

Since the speed of I/O-operations on supercomputers is often slow, writing large amounts of data (such as control points that include several 3D-arrays) can take unacceptably long time. In case of frequent data dumps or slow file system, the time of calculations could be even comparable to the time of I/O, thus it is very important to optimize interaction with the file system. Its realization can be synchronous (blocking) or asynchronous (non-blocking) (see, e.g., (Balaji et al., 2017)).

In the former case, I/O operations are performed by some subset of the processor cores of physical model components, thus inhibiting the physical equations solving.

In the latter case, this inhibition is avoided at the cost of allotting distinct cores to specific I/O services and making procedures for data transfer between these services and the physical components. It is worth noting that increase in the number of writing

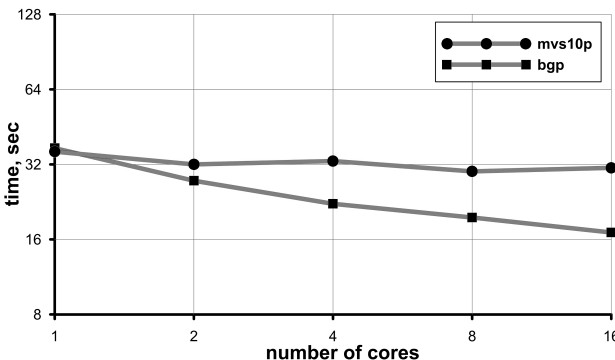

**Figure 5.** Walltime of parallel writing of a model array of $4096 \times 2048 \times 50$ size by different numbers of CMF2.0 coupler cores on MVS-10P and BlueGene/P supercomputers.

cores does not always increase the recording speed, but often reduces it. The particular behaviour is defined by actually installed supercomputer hardware. For example, presence of a single I/O channel for the whole machine can serialize the I/O, and, in opposite, multiple special I/O-nodes may allow one to achieve even some acceleration. But, even in the case of slow hardware, the total time of model experiment can be equal to the time of physical equations solving due to overlapping of computations and I/O. The scheme allows a model to accelerate as long as the writing time is less than or equal to the time of performing

the chunk of calculations. This limitation is controlled by the hardware bandwidth and by the model-service data transfer realisation.

    The asynchronous approach is generally more flexible, so it was chosen for the CMF. In the CMF2.0, all I/O actions are performed by the coupler. The realization is fully parallel, so one can work with any grid sizes just increasing the number of coupler cores. Test results of the CMF2.0 I/O system for the case of writing a single-precision model array of $4096 \times 2048 \times 50$

size are shown on Fig. 5.

    It can be seen that the writing speed of MVS-10P is approximately constant. Moreover, the timing does not change when writing cores are allotted on one or several nodes. The reason is that MVS-10P has only one I/O node and all file operations are performed through it. On the other hand, the BlueGene/P system has several I/O nodes, so a reduction in the writing time is obtained when increasing the number of coupler cores. Nevertheless, the main advantage of the CMF I/O system is

its asynchrony and memory scalability. The acceleration obtained on the BlueGene/P system is rather a nice particular result than a permanent CMF feature. It draws attention to the need of developing I/O infrastructure on supercomputer systems. It's obvious that scalability graphs of a future exaflops machine with millions of cores become very artificial if one has to work with the file system through a single channel.

### 3.5   Additional features

Apart from the coupler, the framework also includes two helpful blocks. For the pre-run stage, the CMF2.0 has got the off-line block, which constructs SCRIP interpolation weights and prepares initial condition files. Like the run-stage CMF program, it

is implemented in terms of abstract operations, which reduces all model configuration actions required from the user (e.g., grid definition) to realization of a few abstract interfaces in a user-derived class.

At the run stage, the user can call various utility modules, like the HaloUpdater, which is needed in finite-difference models. It uses 4-neighbour scheme of any length/dimension/type update on latitude-longitude and tripolar grids, still handling diagonal cells. Impact of the HaloUpdater on performance of the INMIO WOM is described later.

Also the CMF2.0 provides helpful tools for automatic building of various model combinations, makefile and skeleton class generation, data preprocessing, and for other infrastructure actions.

## 4 CMF3.0

### 4.1 PGAS abstraction

The CMF2.0 has shown itself as a suitable framework for high-resolution coupled modeling, allowing us to perform long-
term experiments which would be impossible without it. But the CMF2.0 still has several points for improvement. First of all, although the pure MPI-based messaging is quite fast, it needs explicit work with sending and receiving buffers. Additionally, development of nested regional models becomes quite difficult using only MPI-routines. The CMF2.0 test results showed that we can easily sacrifice some performance and choose better (but perhaps less computationally efficient) abstraction to simplify messaging routines.

We have chosen the Global Arrays library (GA), which implements the Partitioned Global Address Space (PGAS) paradigm of parallel communication and provides an interface that allows to distribute data while maintaining the type of global index space and programming syntax similar to that available when programming on a single processor (Nieplocha et al., 2006). The general idea is to give the user easy access to different parts of a distributed array. The PGAS abstraction assumes that there is a virtual huge array, which is accessible from any process involved in its creation. In fact, there is no global array, and its parts
are stored locally in the memory of processes. But the user does not know about it, since the library takes over all the details, due to which the simplicity is achieved. For example, the client on process X may request an array element with indexes [i, j], as if it has direct access to it. Behind the scenes, GA learns which process holds this element (process Y), executes MPI send-receive transfer and returns the result to the client.

Development of this idea in the CMF3.0 has resulted in the class Communicator_GA, which encapsulates the logic of work-
ing with the GA and provides an interface for put/get operations of sections of global arrays from different model components and services. Moreover, this interface could be used not only for connections between models (including nested ones), but also as a communication mechanism between the models and the coupler, because it allows one to hide all decomposition-to-decomposition problems rising in distributed-memory applications. In the CMF3.0, every array, which participates in intermodel communications, has its "mirror" in the corresponding virtual global array. When the model needs to perform some action, it puts/gets data to/from the global array (this operation is local since the global arrays' processor-wise allocation perfectly matches the model decomposition) and continues calculations. Service components get the array from *the other side*, but

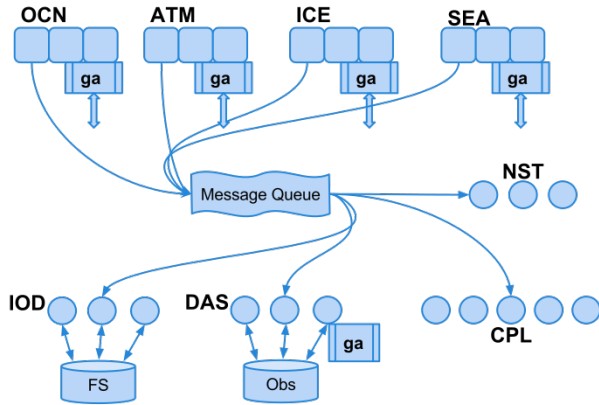

**Figure 6.** The architecture of the compact framework CMF3.0. There are four components in this example: ocean model (OCN), ice model (ICE), atmosphere model (ATM) and sea model (SEA). The components send requests to the common message queue, from where they are retrieved by the coupler (CPL), data assimilation (DAS), input and output data (IOD), and nesting (NST) services. The data itself is transferred through the mechanism of global arrays, which is also used for interprocessor communications in the components and services.

5  this time on their own decompositions. For example, the ocean component could store a global array on 5000 cores with some 2D decomposition, while the I/O procedure outputting this array could utilize only 4 cores and a 1D decomposition.

## 4.2 SOA-architecture

As the complexity of coupled models is growing, we need an easy and convenient way of connecting model components together. The SOA, which was originally introduced for web applications, gives a good pattern for component interactions.
10  In the CMF3.0, all model components send their requests to the common message queue. The service components receive only the messages they could process, then get data from appropriate global arrays and perform the required actions. Such architecture allows us to minimize dependencies between physical and service components, and makes development much easier. Moreover, since all services in the CMF3.0 are based on the same template (inherit the base class Service), it also allows the user to easily add new services to the system by filling only few abstract interfaces. Today, we have four completely
15  independent services built into the CMF3.0: CPL (mapping), IOD (I/O service), NST (nesting service), DAS (data assimilation service).

The CPL service represents the coupler from the CMF2.0 and serves all mapping requests. It receives data through the Communicator_GA class routines, performs interpolation and pushes data to the destination global array (without a request from the receiving side). Although the central coupler architecture of CMF2.0 allows one to collect all service operations on one external component and to perform each of them in parallel, simultaneous requests sometimes can lead to inefficient usage

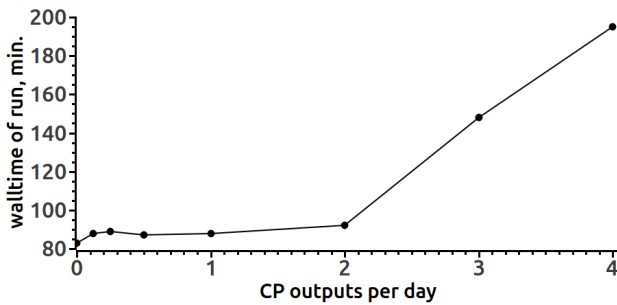

**Figure 7.** Walltime of 8-day run of INMIO World ocean model with 0.1°resolution vs. frequency of saving solution control points.

of processor time. For example, the CMF2.0 coupler can not perform parallel mapping and parallel I/O operations together. This is a disadvantage of all data transfer schemes that combine two or more actions on one process.

In the CMF3.0, we decided to pick out a separate I/O-service, responsible only for working with the file system. For example, when writing data to a file, on the component side it works as follows: the model component has to wait till the corresponding GA-array is free; put the data into the GA-array; mark the GA-array as full; send request for the IOD service. On the IOD side the request is read by the service; the service then takes the requested data from the GA-array; marks the array as free; calls NetCDF routines (same as in the CMF2.0) for parallel writing to file. This approach, though not expected to perform faster than the CMF2.0 direct MPI messaging, provides a flexible and fully asynchronous data writing, limited mainly by the bandwidth of the file system.

A performance test of the CMF3.0 I/O system in the INMIO World ocean model of 0.1°resolution is shown on Fig. 7. In this test we compare the walltimes of 8-day ocean model runs (full physics equations solving) on 600 ocean and 10 IOD cores of MVS-10P system with different frequencies of solution control point (CP) writing to file. The frequencies range from one CP per 6 model hours to one CP per 8 days, the case of no output is also examined. One CP output (file of size 8 GB) takes about 5 wallclock minutes, while one model day physics calculations take about 10 minutes, so one may expect that overlapping of computation and output will be possible if control points are written not more often than twice a model day. Indeed, we see that the graph shows linear growing of required run time if output frequency is greater than 2 CP's per day. This is a quite satisfactory value for long-time experiments such as Earth climate studies (e.g., (Marzocchi et al., 2015), where (1/12)°global ocean model outputs are stored as successive 5-day means).

It should be noted, that one external I/O-service solves only part of the problem, because in case of writing large control point files and dumping frequent lightweight diagnostics the model still would be blocked by the former. Therefore, the service may be further split into two parts – fast and slow I/O-devices. Due to the abstract structure of the Service class this separation can be done via few lines of code.

Further development of the CMF has included data assimilation algorithms. For the ocean model, we have added the new DAS-service, which implements the logic of parallel data assimilation (Kaurkin et al., 2016a), (Kaurkin et al., 2016b).

## 4.3 Class Communicator_GA

The Communicator_GA is a CMF3.0 system class that represents a kind of facade for the GA library. That is, it defines a high-level interface and hides some subtleties of the GA from the user. For example, the class allows to create an array that will be distributed on one component, but still visible to another component. It can be a temperature array that is physically distributed over the ocean's cores (and they can read and write data to it), but, in addition, the CPL service can also work with this array, although it does not store any part of it. Creation of such global array in CMF3.0 will require just a few subroutine calls:

- request the CMF system for component identificators and process lists of currently running ocean model and CPL service;

- register this joint group of processes, prescribing ocean as the holder and CPL as the subscriber;

- request the system for current ocean decomposition;

- register the array, specifying the ocean as the holder and passing its decomposition (so that GA distributes the mirror array in the same way as the model component does with the original one), and the coupler as the subscriber.

The GA put and get operations now may be called. For the holder side they will be local due to consistency of the decomposition. One of the benefits of this architecture is that now the model decomposition can be arbitrary. For example, it becomes easy not to reserve processor cores for subdomains of an ocean model that lay on land.

Every put/get operation must maintain explicit synchronization by setting the array status accordingly to "full" or "empty". This is required since we are not allowed to "lose data". That is, even if some component (e.g., ocean model) is faster than another component (atmosphere model, or IOD in case of too frequent data dumps), we must not lose an array. Accumulation of arrays in a queue also will not lead to success, since models usually work at constant speeds and, as a result, the queue will soon exhaust all available memory. So, if the "fast" model is ready to put/get data, but the GA array is still occupied/empty, the model is blocked.

## 4.4 Interpolation

Since the logic of interpolation subroutines in the CMF3.0 remains the same as in the CMF2.0, we can greatly simplify it by use of GA abstractions. Now, all source data needed by the destination cell is collected directly by Communicator_GA routines. The optimizations regarding repeated cells are preserved. Disadvantage of using the GA is a decrease in performance, since it can not provide persistent operations, overlapping of computations and communication in one service, and obviously has its own overheads. We take the same parameters and input files as of the Test I to compare the CMF3.0 performance with that of the CMF2.0 (Fig. 8). Again, we measure the overall timing including costs of sending event request, sending data, interpolation process and pushing data into destination arrays. Therefore, this timing reflects the overall system overhead against the timing of physical model components.

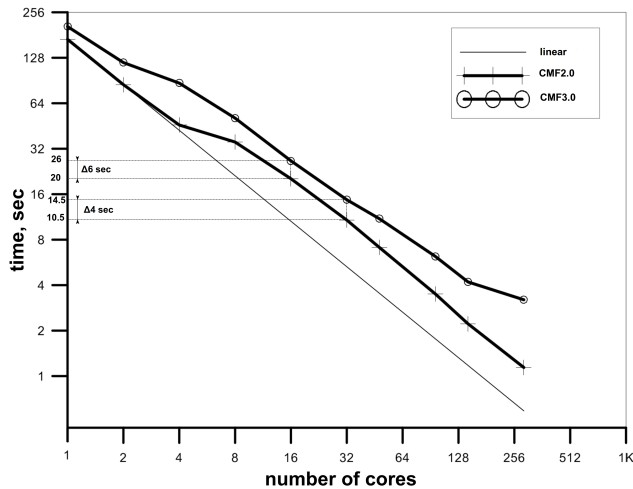

**Figure 8.** Walltime required for the 10-day ocean-atmosphere model run with disabled physics vs. number of coupler cores on the MVS-10P supercomputer (Test I for CMF2.0 and CMF3.0).

Tests were conducted on the MVS-10P supercomputer configuration with 16 cores per node. The graph shows that results are not as good as those for the CMF2.0 (Fig. 2 curve mvs10p_16 is repeated here), but the linear scalability trend is preserved. Moreover, the rate of 2-3 seconds per modeling day (on 20-50 CPL cores) is quite satisfactory for our practical purposes in high-resolution numerical experiments. The decline in performance is expected due to the overhead of using GA-arrays (as an intermediate send/receive data representation) and due to deprecated MPI_SEND_INIT procedures in the CMF3.0. This is a sacrifice for the compact code representation and for convenience of adding new features.

### 4.5  Additional features

At any moment during the run time, the CMF3.0 services can respond to request messages and trigger certain actions on data arrays. So, the model is allowed to send such requests (i.e. raise events) unexpectedly, at any step of its time cycle. Nevertheless, sometimes we may know in advance a schedule of actions (e.g., sending mapping every 2 hours, diagnostics every day and control point every month). The CMF3.0 provides a simple mechanism for generation of such scheduled actions in order to save the user from having to keep track of time and send requests at the right moments. At present, we have two types of event generators: NormalEvent, which represents uniform actions (like diagnostics saving, etc.), and SyncVarEvent, which allows one to synchronize with the time axis of a NetCDF-file (it is useful for experiments with prescribed forcing referenced to the real calendar, e.g. the Drakkar Forcing Set (Dussin et al., 2016)). Depending on current model time, these objects send requests or do nothing. Generators are realized by an abstract class, so for new specific events generator subclasses could be easily added.

In case of unexpected behaviour (like exceptions in model physics or changes in external data) the user can directly call the raise event routine, e.g., for emergency data dump or even to change the functioning of other model components by special messages.

The first to respond to an event is the model component itself – it looks at the event's type and determines what to do (e.g., in the case of saving diagnostics: put the data into the GA-array, mark it as full, send a request to the services, and continue running). Then, the event is packed into an MPI-message and sent as a request to all services (if the model has decided to send it). Services unpack the event, look at its name, and either process it, or ignore.

Other parallel utilities implemented in the CMF3.0 include

- Array operations, such as resizing, changing index order, converting to string and back, search for a particular element;

- Calculating global sums and area integrals over a decomposed model field, which is important in maintaining conservation in geophysical fluid models (e.g., to correct the precipitation, evaporation and runoff algebraic sum in stand-alone ocean simulations like (Griffies et al., 2012));

- Memory usage monitoring.

In the CMF3.0, we included all pre- and post-processing utility modules available in CMF2.0. It is not difficult to migrate from the CMF2.0 to the CMF3.0. Only one adapter file, about 200 lines of code, should be rewritten. It contains several procedures (*ini_main, make_step, finalize*, etc), besides defining events and arrays (intended for I/O, remapping, etc.) registered in it.

## 5   CMF examples of usage

There are several examples of using CMF for various geophysical numerical models:

1. Eddy-resolving ocean dynamics modeling using the INMIO WOM with 0.1°resolution, governed by the CMF2.0 (Ibrayev et al., 2012),(Ushakov and Ibrayev, 2018).

2. Data assimilation of satellite observations and ARGO floats measurements using the DAS service in forecast and reanalysis experiments with the INMIO WOM governed by the CMF3.0 (Kaurkin et al., 2016a).

3. There is a set of works with coupled atmosphere-ocean models for climate change research and numerical weather prediction. The SLAV global atmosphere model (Tolstykh et al., 2017) and the INMIO WOM (Ibrayev et al., 2012) were coupled using the CMF2.0 and CMF3.0 (Fadeev et al., 2016). The results of numerical experiments with the coupled model demonstrate agreement with observational data and show a possibility to use this model for probabilistic weather forecasts at time scales from weeks to year.

4. The nesting technology implemented in the CMF3.0 NST service has been tested for the local INMIO-based model of the Barents Sea with a resolution of 0.1°and the INMIO WOM with a resolution of 0.5°with different geophysical parametrizations (Koromyslov et al., 2017).

5. First results of the seasonal variability simulation for the Arctic and North Atlantic ocean waters and ice by the coupled INMIO WOM and a sea-ice CICE5.1 (Turner and Hunke, 2015) models were obtained under the CMF2.0 in (Ushakov et al., 2016). The numerical experiments were performed in conditions of the CORE-II protocol.

## 5.1 INMIO World Ocean Model

As it was mentioned, one of the goals of the CMF is to provide tools for effective parallel calculations of stand-alone models.
Historically, it was developed to provide efficient support for the INMIO WOM. This model utilizes a 2D-decomposition of the tripolar grid. Increasing the number of cores decreases (almost proportionally) the number of performed operations for each process, since the model uses explicit time schemes for horizontal operators, which require only local halo updates. Therefore, limitations in scalability can only be associated with halo update routines and external blocks (e.g., in the I/O system).

The latest version of INMIO WOM is distributed in an integrated package together with the CMF2.0 and 3.0, all necessary
libraries and a standardized folder structure facilitating the adding of new model components (including adapter files for the CICE sea-ice model). At present, the INMIO code consists of the hydrodynamical solver, atmospheric boundary layer bulk formulae, the built-in thermodynamic ice model of (Schrum and Backhaus, 1999) (turned off in case of coupling with the CICE model), and online data processing routines (e.g., averaging). The intramodel communications (halo exchanges on tripolar and latitude-longitude grids) and work with the file system are delegated to the CMF utilities module. For prescribed experiments
with the CORE (Griffies et al., 2012) forcing, two data models (reading CORE data files) are also registered as separate atmosphere and land components, and the CMF coupler provides interpolation of their fields onto the ocean grid.

Scalability of the INMIO WOM of $0.1°$resolution driven by the CMF2.0 is shown on Fig. 9. Maximum number of Blue-Gene/Q cores utilized is 32400 (32000 for the ocean component and 400 for the coupler). The parallel efficiency of this configuration in relation to the configuration of 8100 cores (8000 for the ocean and 100 for the coupler) is 78 %. Obviously, at
high core counts the parallel efficiency curve experiences some "flattening". But assuming that the time step of the model is 5 min., the result of the experiment leads, e.g., to quite satisfactory five simulated years per wall-clock day (SYPD) rate achieved on 20000 cores of the BlueGene/Q supercomputer.

## 5.2 Coupled Global atmosphere - ocean model

The second application of the framework was the numerical experiment with the global coupled INMIO ocean (Ibrayev et al.,
2012) and SLAV atmosphere (Tolstykh et al., 2017) models. The SLAV model with horizontal resolution $0.9° \times 0.72°$ and 28 vertical levels, and the INMIO WOM with resolution $0.25°$ and 49 vertical levels were coupled into a single program using the CMF2.0 system. The short- and long-wave radiation in the SLAV model were computed with the time-step of 1 hour. The spatio-temporal resolution was restricted by available computer resources but not by the CMF itself.

Prognostic coupled model calculations were carried out with a time step of 6 min. for the oceanic component and 3.6 min.
for the atmospheric one. The initial state of the ocean was obtained by a spin-up of the standalone ocean model driven by the ERA-Interim atmospheric forcing. The atmosphere started from the objective analysis of the Hydrometcenter of Russia. Every 72 min., nine 2D-arrays were transferred from the atmosphere to the ocean (components of wind stress, short- and long-wave

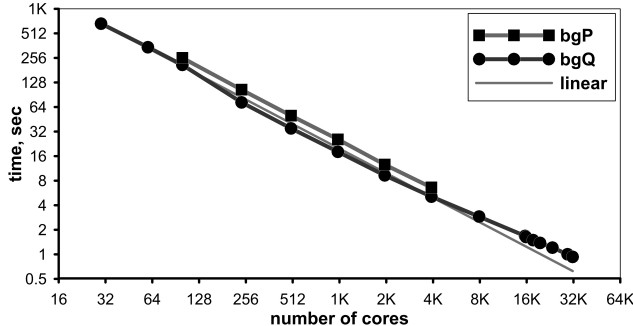

**Figure 9.** Walltime of the 0.1°resolution INMIO WOM (governed by the CMF2.0) 10 model steps vs. model communicator size on the BlueGene/P supercomputer (Moscow State University) and BlueGene/Q supercomputer (IBM Research Center Thomas J. Watson).

radiation, fluxes of sensible and latent heat, precipitation, evaporation, air temperature at 2 m). Conversely, every 144 min. three 2D-arrays were transferred from the ocean to the atmosphere (upper gridbox temperature, temperature and concentration of sea
ice). The sea ice was simulated by the INMIO built-in ice thermodynamics model, while the land processes were incorporated into the SLAV atmosphere model. The coupled model works stably and, along with intraannual distribution characteristics of monthly data fields, reproduces enough thin elements of atmospheric and oceanic circulation.

The model throughput on the MVS-10P supercomputer was equal to 0.75 SYPD for the configuration ocean (1152 cores) – atmosphere (288 cores) – coupler (16 cores). At that moment, the maximal communicator size available for the atmosphere
model was limited due to the one-dimensional latitudinal grid decomposition.

## 5.3 Data assimilation using DAS

As well as any service of the CMF3.0, the data assimilation is performed on separate processor cores. This allows to structure the Earth modeling system better, in order to make each software component solve its own problem. At the same time, the model of the ocean does not take part in the data assimilation. Only results of the ocean modeling in the form of ensemble
vector elements are used. On their basis, the covariance matrices are approximated. The data from the ocean model is sent to the service (usually once a modeling day) without using the file system (through the cluster interconnect). Moreover, all matrix-vector operations are calculated in parallel (on shared memory) using BLAS and LAPACK functions through the Global Arrays (GA) toolkit (Kaurkin et al., 2016a).

Due to the effective implementation of the EnOI method as the DAS parallel software service, the data assimilation problem
scales almost linearly (Fig. 10). So, assimilation of $10^4$ observational points of satellite data on 16 processor cores takes about 20 seconds instead of 5 minutes on a single core, which would be comparable to the time spent on daily ocean model forecast for 200 cores.

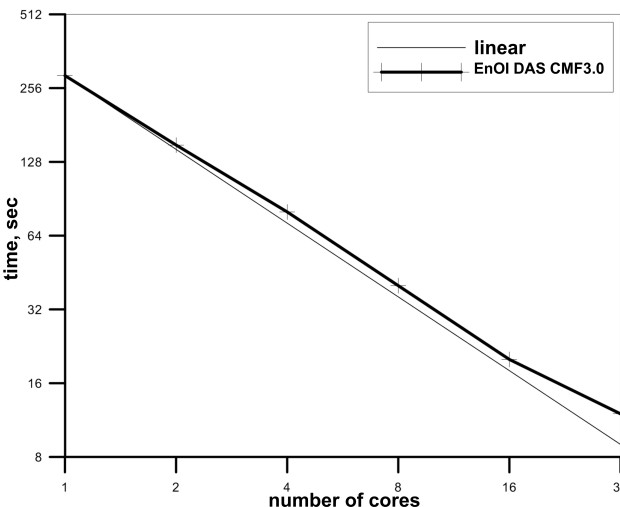

**Figure 10.** Scalability of the EnOI method in context of the CMF3.0 DAS service at the assimilation of $10^4$ points on the Lomonosov supercomputer (Moscow State University).

## 6  Summary and conclusions

We present an original modeling framework CMF developed as our first step to high resolution modeling. The key part of it, the coupler, in the initial version CMF2.0 has a sufficiently small code size for such programs (about 5000 lines of code including unit tests) and is able to manage the main parallel problems of the coupled modeling – synchronization, regridding and I/O. The coupled model follows the single executable design with the main program independent of components' code, and the coupler dealing with all service operations. The new version, CMF3.0, utilizes the SOA-design, which allows one to divide the coupler responsibilities into small separate services, easily plug/unplug them and add new ones to the system, thus providing a further generalization to the coupling interface. The PGAS messaging greatly simplifies implementation of all model low-level interprocess communications.

Tests for CMF2.0 parallel mapping efficiency were carried out on four modern supercomputer architectures. They show a nearly linear scalability of the overall communication system and the regridding procedure. Satisfactory speed results could be achieved already on 20-40 coupler cores even dealing with grids of high resolution (0.1°for the ocean and 0.225°for the atmosphere). I/O tests proved the ability of the coupler asynchronous I/O scheme to handle huge amounts of data. As expected, the new CMF 3.0 version has lower absolute performance, but still preserves linear trend of scalability and suitable timing (2-3 seconds per modeling day on 20-50 coupler cores) for high-resolution modeling. The parallel data output speed of CMF3.0 is about a half of that for CMF2.0, but still it is quite satisfactory for contemporary high-resolution climate experiments due to overlapping of computation and writing to file system. Within the framework of the CMF3.0 architecture it was possible to effectively implement nesting and data assimilation technologies. This functionality is not yet common for other coupling platforms.

Originally designed for the INMIO World ocean model support, the CMF has developed into a flexible and extensible instrument providing means for high-resolution resource-demanding simulations in regional/global, stand-alone/coupled, and forecast/climate problems.

*Code availability.* The code of the CMF3.0 and CMF2.0 (distributed under GPLv2 licence) is available on http://model.ocean.ru (after registration).

## Appendix A: Supercomputer configurations used

The MVS-100k and MVS-10P systems are installed at the Joint Supercomputer Center of the Russian Academy of Sciences (www.jscc.ru). The MVS-100k consists of 1460 modules (11680 processor cores). The basic computing module is an HP Proliant server, containing two quad-core Intel Xeon processors running at 3 GHz on 8 GB RAM. Computational modules are interconnected with Infiniband DDR. The MVS-10P system includes 207 nodes. Each node incorporates 2 Intel Xeon E5-2690 processors (16 cores on 2.90 GHz), 64 GB of RAM, and two Intel Xeon Phi 7110H coprocessors. Computing nodes are combined into an FDR Infiniband network.

Supercomputer BlueGene/P is located at the Faculty of Computational Mathematics and Cybernetics, Moscow State University, and consists of 2048 computing nodes. Each node has four PowerPC 450 cores (850 MHz) and 2 GB of RAM. Nodes are networked with the 3D-torus topology (5.1 GB/s, DMA).

Supercomputer BlueGene/Q is located at the IBM Thomas J. Watson Research Center and consists of several racks. Every two racks have 2048 computational nodes, each with 16 cores. The core is a PowerPC A2 (16 GB RAM, 1.6 GHz). Nodes are networked with the 5D-torus topology (40 GB/s, DMA).

Supercomputer Lomonosov is located at the Lomonosov Moscow State University and consists of more than 50000 cores. We have used the partition with eight-core nodes (2 x Intel Xeon 5570 Nehalem, 12 GB, 2.9 Ghz). Computational modules are interconnected with the Infiniband QDR.

*Acknowledgements.* The research of Sections 1–3, 5.1 and 5.2 was supported by the Russian Science Foundation (project no. 14-37-00053) and performed at the Hydrometeorological Research Center of the Russian Federation. The research of Sections 4 and 5.3 was supported by the Russian Science Foundation (project no. 17-77-30001) and performed at the Federal State Budget Scientific Institution "Marine Hydrophysical Institute of RAS".

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
