# Peer review of "Compact Modeling Framework v3.0 for high-resolution global ocean-ice-atmosphere models"

_Geoscientific Model Development, 2017_

## Referee Comment (RC1) · Anonymous Referee #1 · 13 Mar 2018

Title: Compact Modeling Framework v3.0 for high-resolution global ocean-ice-atmosphere models Author(s): Vladimir V. Kalmykov et al. MS No.: gmd-2017-294

General Comments:

This paper provides an overview of the Compact Modeling Framework (CMF2.0 and CMF3.0) implementation. The paper is well organized. Performance plots are shown for several high resolution cases. It would be nice if the performance plots were extended to higher core counts if possible. The paper could use an additional review by a native English speaker as much of the paper includes some grammatical challenges. Specifically, lack of "a" and "the" in the paper could be much improved.

[Figure]

Specific Comments:

page 2, line 53. Please define WOM at first use and review that definitions exist for other acronyms.

page 4, line 33. Please define SOA at first use and review that definitions exist for other acronyms at first use.

Figure 1 implies that the coupler has distinct cores. Please make sure this is also clearly stated in the text. The picture in Figure 1 suggests there is a 1:1 connection between model tasks and coupler tasks, but this is highly unlikely in practice. It might be clearer if each component had different numbers of tasks in Figure 1. The figure also implies the decomposition on the coupler is the same as the decomposition in the models. But then this does not guarantee "locality of data and communications during the interpolation process or I/O actions" as stated on page 4, line 50. Either the coupler has "near" 1:1 communication with physical models and then interpolation requires a rearrange communication OR there is M:N communication between physical models and the coupler and then minimal communication as part of interpolation. The only way both communication to coupler and interpolation communication can be minimized is if the model decompositions are all chosen very carefully. Again, in practice, this will not be the case. Some rethinking about how this is stated and shown would be helpful.

page 5, line 60. Please provide additional details on how this is implemented.

Figures 2-4 and Figure 6. It would be nice if there were some additional results at higher core counts. I recognize the authors feel this is not needed because the performance of the model is adequate as shown. It still would be informative to the community to see how far the strong scaling goes in their implementation.

Figure 4. The log scaling does not show the detailed information of the relative performance of different cases for a fixed core count. The text notes the percentage differences of a few cases, and this is interesting but incomplete. I wonder if it might

be better to show the data differently, maybe in a table, or maybe in a plot where the y-axis was linear with non-dimensionalized scaling units.

Section 3.4 describes some theoretical ideas about cost for 4 different I/O schemes. It closes by indicating the asynchronous scheme was chosen and that it works without providing any further results. I think, at the least, the performance of the implementation should be documented with actual numbers and then compared with the theoretical description. It would be great if that scheme could be compared to the 3 other schemes, although recognize this might not be possible. The description of the 4 schemes could certainly be reduced, especially as no results are presented for them. Result from the actual performance of the implementation should be increased and described in more detail.

Figure 6, could the CMF2.0 results be added to the plot. This is brought up directly on page 11, line 15 and then again on page 11, line 19.

page 11, line 19. "as expected". Please expand on this, why is it expected?

More generally, please expand on the differences in CMF2.0 and CMF3.0. They both have the coupler on separate cores. CMF3.0 has an additional buffer layer, how is this beneficial, what works well, what doesn't work so well? It is slower than CMF2.0 so how does the community feel about the implementation?

page 13, line 61. Please state how many cores the coupler was using. This should be noted in all application results.

Figure 8. It would be nice if this plot were formatted similar to plots 2-4 with time instead of acceleration on the y-axis, for consistency. Even if it's not log-log and even if it's relative time in this case.

Technical Comments:

I will not go thru each grammatical error but strongly encourage additional review by a native English speaker. Let me just propose an update to the Abstract, for instance,

We present a new version of the Compact Modeling Framework (CMF3.0) developed for the software environment of stand-alone and coupled global geophysical fluid models. The CMF3.0 is designed for use on high and ultra-high resolution models on massively-parallel supercomputers. The key features of the previous CMF version (2.0) are mentioned to reflect progress in our research. In the CMF3.0, the MPI approach with a high-level abstract driver, optimized coupler interpolation, and I/O algorithms is replaced with the PGAS paradigm communications scheme, while the central hub architecture evolves to a set of simultaneously working services. Performance tests for both versions are carried out. In addition, a parallel realisation of the EnOI (Ensemble Optimal Interpolation) data assimilation method as a program service of CMF3.0 is presented.

Much of the document could use similar revision. There are issues throughout.

page 7, line 3, "communicational" is not a word and that sentence makes little sense as written.

page 11, line 11, using -> use

page 14, line 2. Starting the sentence with SYPD is not ideal. Just say "The model throughput" and provide units on the 0.75 value.

page 15, line 27, remove "with" in "handle with huge"

page 15, line 34, change "to further" to "for further"
* * *

---

## Referee Comment (RC2) · Anonymous Referee #2 · 16 Apr 2018

This paper presents versions 2.0 (based on MPI) and 3.0 (based on PGAS) of the Compact Modeling Framework, a software infrastructure offering different services (coupling, I/O, data assimilation) to implement numerical geophysical models. Tests on different architectures are performed and show the scalability of the framework for relatively high-resolution problems. The paper contains interesting information on CMF that would deserve publication in GMD but I consider important parts are missing and the manuscript certainly needs major revisions before it can be considered for publication.

A) Major comments:

A.1) The description of CMF2.0, results of Test I and II and related conclusions need to be clarified:

- First, I agree with Referee #1 that figure 1 is misleading as it suggests a 1:1 connection between the models processes and the coupler processes. I don't understand either the "which means data locality" on P.4, L.50, as there is certainly some exchange of data needed between the component processes and the coupler processes.

- P.5, L.63-66: these two paragraphs are not clear at all. What does "a subset of component's cores works only with individual master core in the coupler" mean? What does Âń for the two cases, of the source and destination type Âż mean and why do you put a reference to Craig et al. 2005 here?

- P.5., L. 69-70: You write "All necessary links are initialized at the beginning of run and are used at the calculation stage as persistent (Jacob et al., 2005). Âż I suppose that the links you mention here are not the SCRIP links? You should be using another word as this is confusing. Also, why do you mention Jacob et al. 2005 here?

- P.5, L.72-73: I don't understand this sentence: "It is worth noting, that links are not sent directly, but as sorted unique cells vectors which allow one to avoid sending duplicated data. Âż. Again what "links" are you talking about here?

- In Figure 2, 3 & 4 captions and on P.6, L.88, you should recall what is included in the timing. I suppose this is the time in seconds for the whole 10 model days for the whole ocean-atmosphere and atmosphere-ocean exchanges through the coupler. (Same remark for Figure 6).

- On p.6, L.92, you write: "It is clear that 20-40 coupler cores provide satisfactory speed for such problems, because ~10 seconds costs . . ." It is not clear how you get these numbers. I see at best, i.e. with the MVS-10p_16, something between ~15 and ~20 seconds. Same remark for the number presented on P.7, L.4-5. This re-joins referee #1's comment about the fact that the log scale does not allow one to get the detailed

information mentioned in the text.

- P.7, L.6: again here I don't understand why you write "perform only local communication"

- Section 3.4 on I/O should be completely revised. The long theoretical section on P.8 with detailed formula is not useful here, especially as you finally simply state "Asynchronous scheme was incorporated in the latest version our framework Âż without giving finally any results! The theoretical section should be cut and numbers obtained for the grid sizes you list for INMIO World ocean model should be provided.

- The different utility modules available should be briefly described or a reference to a documentation or User Guide should be provided.

A.2) The description of CMF3.0 needs to be extended

This paper is supposed to mainly describe CMF3.0, or at least this is what the title implies but very little is written about it. It looks like the author was in a hurry to finish the paper. More details should certainly be given on how the I/O service work (1st paragraph on P.11) and on the Data Assimilation part (currently only 2 lines, P.11, L8-9). These improved descriptions should be backed up with performance results (as is done for the interpolation).

Also the discussion of the interpolation results should be extended and detailed. How do the author get to 2-3 seconds per modelling day on 20-50 CPL cores? (This is mentioned also in the conclusions P.15, L.28.) Why is it expected that results would be worse than for CMF2.0 (and "worse" should not be used here because it implies that results for CMF2.0 are bad and that results for CMF3.0 are even worse)? Is it because of the shift from MPI to PGAS? Or because the tests were performed on a different platform?

Also, in the conclusions, you write: "The key part of it, coupler, has a sufficiently small code size for such programs (about 5000 lines of code with unit tests) and is able to

manage the main parallel problems of the coupled modeling - synchronization, regridding and I/O." I don't understand why you write that the coupler manages the I/O as this is not the case in CMF3.0.

A.3) The whole text needs reviewing by a native English speaker. The style and wording needs revision as some sentences are simply not understandable (at least by me), e.g.:

- P.3, L.16: "Unquestionable advantage of non-coupler design is the absence of interference in the user code": why "non-coupler design"? Also, with OASIS, there is some interference in the user code but the objective is to minimize it.

- P.5, L.76: "Several ping-pong tests were carried out for interpolation system using coupled ocean-atmosphere model."

- P.6, L.95-96: "work of the sequential algorithm is only possible with restriction that memory is allocated only for interpolation block, which is impossible in practice Âż: I am not sure what this sentence means exactly and why you write that it is impossible in practice while you do get some results on 1 core; do you mean it would be impossible with real models as the interpolation per se would require all the available memory?

- P.13, L.63: "but we are more interested in scalability of the program on perspective sizes of computational resources Âż

- P. 13, L.71: "Time evolution of the sea-ice surface temperature is described in the same way as in prescribed ocean experiments.":

- P. 13, last paragraph: please rephrase, the current sentence with the "min." is too difficult to follow.

- P.2, L25: "Coupling through shared file or sequential component is acceptable . . . Âż could be Âń Coupling through shared file with components executing sequentially is acceptable . . . Âż

- P.2, L33: "... and their representation in the interfaced style understandable ..." could be " ... and their adaptation to the interface understandable ..."

- P.2, L41: "GFDL FMS (Balaji, 2012) system additionally suggests fully parallel data storage with file post processing at the end of the run Âż could be Âńln the GFDL FMS (Balaji, 2012) system, fully parallel data storage with file post processing at the end of the run is offered Âż

- P.3, L.21: "According to proposals of Earth System Modeling conference, (Valcke et al., 2012), ..." could be "According to the analysis of coupling technologies for Earth System Modeling by Valcke et al. 2012, ..."

- P.6, L86: "Performance is based on a standard Intel Fortran compiler" should be moved to the next paragraph and could be "On all supercomputers, the coupled system was compiled standard Intel Fortran compiler."

- P.6, L.90: "Increasing number of coupler size"; should be "Increasing number of coupler processes" or "Increasing the size of the coupler communicator"

- P. 11, L. 12: "Optimizations regarded to ignore repeated cell requests are preserved" could be "Optimization regarding repeated cells are preserved".

- P. 13, L.56: "Latest version of INMIOWOM model was fully integrated to CMF Âż could be Âń CMF2.0 was fully integrated in the latest version of INMIOWOM Âż

- P.14, 3rd line: "Ice model was built into the ocean model , land model – into the atmosphere model" could be "The ice model is integrated in the ocean model and the land model in the atmosphere model"

- P.14, L.6: "structurize" could be "structure"

B) Other comments:

B.1) References:

- P.2, L.6: The reference to OASIS3 should be Valcke 2013 (i.e. p18, L45)

- P.2, L.36: The reference to OASIS3-MCT should be Craig et al 2017: A. Craig, S. Valcke, L. Coquart, 2017: Development and performance of a new version of the OASIS coupler, OASIS3-MCT_3.0, Geosci. Model Dev., 10, 3297-3308, https://doi.org/10.5194/gmd-10-3297-2017, 2017.

- P.3, L.7 (as annotated in the manuscript): Reference to ESMF should be the more recent one: Theurich, G., Deluca, C., Campbell, T., Liu, F., Saint, K., Verten- stein, M., Chen, J., Oehmke, R., Doyle, J., Whitcomb, T., Wall- craft, A., Iredell, M., Black, T., Da Silva, A. M., Clune, T., Fer- raro, R., Li, P., Kelley, M., Aleinov, I., Balaji, V., Zadeh, N., Ja- cob, R., Kirtman, B., Giraldo, F., McCarren, D., Sandgathe, S., Peckham, S., and Dunlap IV, R.: The Earth System Prediction Suite: Toward a Coordinated U.S. Modeling Capability, B. Am. Meteor. Soc., 97, 1229–1247, https://doi.org/10.1175/BAMS-D- 14-00164.1, 2016.

B.2) P.2, L.35-36: OASIS3-MCT proposes coupled system not only as single exe- cutable, so this sentence is misleading; the important feature is that the coupling func- tions are not provided by a standalone coupler but by a coupling library linked to the component models. Please correct.

B.3) P.2, L.44-50: This list mixes functionalities (1. and 2.) and characteristics (3. and 4.); please reorganise.

B.4) P.3, L.11: It is not really fair to write that the computational costs of CESM coupler are quite significant 20%, as the CESM coupler does not only perform coupling and remapping but also performs the surface flux computation.

B.5) P.3, L. 21-24: these 3 lines do not give an appropriate summary of the analysis provided by Valcke et al. 2012. Please correct.

B.6) P.4, L.42-43 & P.9, L.47: Either provide details on what interceptor or Template methods are or don't mention them; one should not have to read the reference (Gamma

et al., 1995) to understand the sentence.

B.7) P.5, L77: References here are misleading as they seem to imply that "Test I condition" are explicitly defined in Valcke et al., 2012 or in Craig et al. 2012, while they are not.

B.8) P.6, L.93: You should not use the word "failure" here as the test does not fail, it is just very slow.

B.9) Using very technical coding terms along the text does not help understanding it (e.g. P.4, L.45: "Component class"; P.9, L.63:"resulted in class Communicator"; p.10, L.77-78:"since all services in CMF3.0 inherit base class Service it also allows one to easily add new Âż ; P.10, L.80 : Âńlt receives data using Communicator Âż ; P.12, L.25 : Âń NormalEvent Âż or Âń SyncVarEvent Âż : P.12, L.27 : ÂńGenerators realize abstract class EventGenerator, so new specific generator subclasses could be easily added ); I think it would be better to explain the concept that using these abstract terms.

B.10) P.12, L.26: Reference to Griffies et al. is not useful here.

B.11) P.12, section 5.1: What is the resolution of the INMIO World Ocean model in these tests?

B.12) P.15, L.25-27: please specify if these numbers apply to CMF2.0 or CMF3.0. On line 27, change "CPL3.0" for "CMF3.0"!

B.13) In general, I think the section 6 on Conclusions and future work could be fleshed out.

B.14) I think the "Code availability" section is not satisfactory regarding GMD standard but I will let the Topical Editor decide on this point.

C) Minor comments:

C.1) In the abstract, you write "As addition a parallel realisation of the EnOI (Ensemble Optimal Interpolation) data assimilation method as program service of CMF3.0 is

presented." but this is not the only example presented in section 5.

C.2) P.3, L.13: It is not right to write that OASIS3 is the most popular version of OASIS as most groups are using OASIS3-MCT today.

C.3) P.3, L.16: In OASIS3-MCT, MCT procedures are executed on all component model cores

C.4) P.4, L.33: Define SOA the first time it appears in the text

C.5) P.2, L53: Define WOM the first time it appears in the text.

---

## Author Comment (AC1) · 26 Jun 2018

Maxim Kaurkin

**maksim.kaurkin@phystech.edu**

Dear Reviewer, thank you very much for finding time to read our article and present your comments that helped to improve the manuscript. The paper has undergone a thorough reformulation, some sections have been extended. Please find our responses below.

Author responses to Reviewer 1 comments

The following is the point-by-point response to the reviewers' comments (shown in "italic").

> ***Referee comment.***
>
> *General Comments:*
>
> *This paper provides an overview of the Compact Modeling Framework (CMF2.0 and CMF3.0) implementation. The paper is well organized. Performance plots are shown for several high resolution cases. It would be nice if the performance plots were extended to higher core counts if possible. The paper could use an additional review by a native English speaker as much of the paper includes some grammatical challenges. Specifically, lack of "a" and "the" in the paper could be much improved.*

**Author's response.**

We have revised the English wording and sentence structure throughout the paper. Two figures and one section have been added, so the numbering has changed. As for higher core counts, please see the answer to comment (4).

> ***Referee comment.***
>
> *Specific Comments:*
>
> *(1) page 2, line 53. Please define WOM at first use and review that definitions exist for other acronyms.*
>
> *page 4, line 33. Please define SOA at first use and review that definitions exist for other acronyms at first use.*

**Author's response.** The terms «World ocean model (WOM)» and «service-oriented architecture (SOA)» have been added to the text (in the beginning and at the end of Section 2, respectively). All other acronyms have been checked.

> ***Referee comment.***

*(2) Figure 1 implies that the coupler has distinct cores. Please make sure this is also clearly stated in the text. The picture in Figure 1 suggests there is a 1:1 connection between model tasks and coupler tasks, but this is highly unlikely in practice. It might be clearer if each component had different numbers of tasks in Figure 1. The figure also implies the decomposition on the coupler is the same as the decomposition in the models. But then this does not guarantee "locality of data and communications during the interpolation process or I/O actions" as stated on page 4, line 50. Either the coupler has "near" 1:1 communication with physical models and then interpolation requires a rearrange communication OR there is M:N communication between physical models and the coupler and then minimal communication as part of interpolation. The only way both communication to coupler and interpolation communication can be minimized is if the model decompositions are all chosen very carefully. Again, in practice, this will not be the case. Some rethinking about how this is stated and shown would be helpful.*

**Author's response.**

The distinction of cores has been explicitly stated in the beginning of the Section 3.1.

Figure 1 has been reworked to show the more general case. It illustrates now the basic idea that every coupler core communicates with a fixed subset of each component's cores.

If we correctly understood the comment of the Reviewer, our work is an attempt to implement particularly the latter case, which optimizes both communication to coupler and interpolation communication. This is achieved as follows:

a) The size of coupler communicator is taken much smaller than the size of any model component communicator, so the amount of rearrange interpolation communications is small (though nonzero).

(b) Each coupler core ("master") interacts only with a specific subset of each component's cores ("slaves"). This allows to reduce the required amount of coupling communication routes for every component from M*N (*which would be in the general case of the component and the coupler running on M and N cores, respectively*) to only M, since each slave now interacts just with its master. Therefore, the size of every component communicator has to be a multiple of the coupler communicator size. In order to meet this condition in practice (e.g., for "poorly divisible'" component grid sizes), the CMF2.0 partially supports uneven grid subdomain sizes by making the last row of the component's 1D- or 2D-decomposition narrower than the others.

We have described this master-slave architecture in the Section 3.2 and removed the term "locality" from the text.

As for the CMF3.0, all communications are carried out through the GA layer, so these optimizations are not applicable. But, on the other hand, the decompositions now can be arbitrary. In particular, it becomes easy not to reserve processor cores for subdomains of a global ocean model that lay on continents.

*Referee comment.*

*(3) "Combination of predefined time chain, persistent communications and pointer based asynchronous data sending provides maximal efficiency of data gathering and distribution."*

*page 5, line 60. Please provide additional details on how this is implemented.*

**Author's response.** We have extended the corresponding paragraphs:

All events in the system are divided into few classes (save diagnostics, save control point, read file data, send/receive mapping, etc.), defining different actions with data arrays. In the CMF2.0, we postulate that all events could be predefined before the start and occur with fixed periods. Thus, the coupler can take on the task of synchronizing models and avoiding deadlocks.

The sequence of events (time chain) is constructed in the main CMF program, which is the entry point of the coupled model. Also, at the registration stage, models provide the CMF system with pointers to the arrays that must be processed in the events. So, during the system operation, events are performed automatically and do not require explicit calls from the user. As the information about the periods of all events is known at the registration stage, the coupler can build a table of its actions. This allows to exclude parallel synchronization of the coupler cores, which otherwise would be necessary when, for example, two components at the same time want to write data to the file system. When a certain time moment arrives, the coupler selects the next event from the chain and calls the appropriate handler function based on the type of this event, while the model components asynchronously send data. Moreover, it becomes possible to use persistent MPI-operations (combinations of MPI_SEND_INIT and MPI_STARTALL) for all events, thus saving time of repeated communications. Combination of predefined time chain, persistent communications and pointer-based asynchronous sending provides high efficiency of parallel data gathering and distribution.

> ***Referee comment.***
>
> *(4) Figures 2-4 and Figure 6. It would be nice if there were some additional results at higher core counts. I recognize the authors feel this is not needed because the performance of the model is adequate as shown. It still would be informative to the community to see how far the strong scaling goes in their implementation.*

**Author's response.**

These performance tests were conducted prior to year 2016 and, unfortunately, due to various reasons there is no access to most supercomputer configurations mentioned in the paper. Particularly, we have no access to the BlueGene/P, BlueGene/Q and Lomonosov, while the MVS100k and MVS10p have been reconfigured which does not allow to continue experiments in the same conditions. Nevertheless, results of the BlueGene tests are most interesting since they are obtained on highest numbers of cores (up to 32K). Moreover, the BlueGene system has the most advanced interconnect, which allowed to achieve almost linear scalability. Thus, these results are left unchanged in the paper as an evidence that the CMF2.0 system can show good results on a high-performance supercomputer.

An improvement was possible for the Figure 6 (now Figure 8), where we changed CMF3.0 Lomonosov results to MVS10p ones, which extend to 256 coupler cores. This also made it possible to compare them with CMF2.0 results obtained in the same conditions, as asked in the Comment (7).

> ***Referee comment.***
>
> *(5) Figure 4. The log scaling does not show the detailed information of the relative performance of different cases for a fixed core count. The text notes the percentage*

*differences of a few cases, and this is interesting but incomplete. I wonder if it might be better to show the data differently, maybe in a table, or maybe in a plot where the y-axis was linear with non-dimensionalized scaling units.*

**Author's response.**

In our opinion, the logarithmic scale with time vs. number of cores is preferable for the analysis of parallel efficiency, because it allows simultaneously to compare with linear trend (which illustrates "perfect scaling") and to know the time of execution (in contrast to a speed-up plot or a non-logarithmic time-cores plot). For convenience of estimations, we have added projection lines to the plots on the Figures 2 and 8. The data of Figure 4 additionally has been presented in a table.

*Referee comment.*

*(6) Section 3.4 describes some theoretical ideas about cost for 4 different I/O schemes. It closes by indicating the asynchronous scheme was chosen and that it works without providing any further results. I think, at the least, the performance of the implementation should be documented with actual numbers and then compared with the theoretical description. It would be great if that scheme could be compared to the 3 other schemes, although recognize this might not be possible. The description of the 4 schemes could certainly be reduced, especially as no results are presented for them. Result from the actual performance of the implementation should be increased and described in more detail.*

We have reduced and rewritten this section, so that it is now devoted to the CMF asynchronous scheme only. An inspection of the other 3 schemes would require a big separate study, so we do not consider them here. Data writing speed test results have been added for the CMF2.0 on the MVS-10P and BlueGene/P systems. In Section 4.2 we have also added and discussed results of the actual asynchrony test of the CMF3.0 I/O system in conditions of the 0.1-degree global ocean model.

*Referee comment.*

*(7) Figure 6, could the CMF2.0 results be added to the plot. This is brought up directly on page 11, line 15 and then again on page 11, line 19.*

**Author's response.**

CMF2.0 results have been added to this figure (now it is Figure 8).

*Referee comment.*

*(8) page 11, line 19. "as expected". Please expand on this, why is it expected?*

*More generally, please expand on the differences in CMF2.0 and CMF3.0. They both have the coupler on separate cores. CMF3.0 has an additional buffer layer, how is this beneficial, what works well, what doesn't work so well? It is slower than CMF2.0 so how does the community feel about the implementation?*

**Author's response.**

The decline in performance is expected due to the overhead of using GA-library (as an intermediate send/recv data representation) and due to deprecated MPI_SEND_INIT procedures in the CMF3.0. This is a sacrifice for the compact code representation, easy parallel messaging, and for convenience of adding new user-defined services. We have expanded the Section 4 in order to clarify these features of the CMF3.0 system. This is one of the first papers about the CMF3.0, so we hope that it will start bringing the feedback from the wide community.

> *Referee comment.*
>
> *(9) page 13, line 61. Please state how many cores the coupler was using. This should be noted in all application results.*

**Author's response.**

The two configurations compared in this section used (32000 ocean cores + 400 coupler cores) and (8000 ocean cores + 100 coupler cores). This has been added to the text.

In Section 5.2 the core counts are (1152 ocean cores + 288 atmopsphere cores + 16 coupler cores).

In Section 5.3 the timing of CMF3.0 DAS service is tested up to 32 cores. The CPL service is not used here.

> *Referee comment.*
>
> *(10) Figure 8. It would be nice if this plot were formatted similar to plots 2-4 with time instead of acceleration on the y-axis, for consistency. Even if it's not log-log and even if it's relative time in this case.*

**Author's response.**

This figure (now Figure 10) has been formatted similar to Figures 2-4.

> *Referee comment.*
>
> *Technical Comments: I will not go thru each grammatical error but strongly encourage additional review by a native English speaker. Let me just propose an update to the Abstract, for instance,*
>
> *We present a new version of the Compact Modeling Framework (CMF3.0) developed for the software environment of stand-alone and coupled global geophysical fluid models. The CMF3.0 is designed for use on high and ultra-high resolution models on massively-parallel supercomputers. The key features of the previous CMF version (2.0) are mentioned to reflect progress in our research. In the CMF3.0, the MPI approach with a high-level abstract driver, optimized coupler interpolation, and I/O algorithms is replaced with the PGAS paradigm communications scheme, while the central hub architecture evolves to a set of simultaneously working services. Performance tests for both versions are carried out.*

*In addition, a parallel realisation of the EnOI (Ensemble Optimal Interpolation) data assimilation method as a program service of CMF3.0 is presented.*

*Much of the document could use similar revision. There are issues throughout.*

*page 7, line 3, "communicational" is not a word and that sentence makes little sense as written.*

*page 11, line 11, using -> use*

*page 14, line 2. Starting the sentence with SYPD is not ideal. Just say "The model throughput" and provide units on the 0.75 value.*

*page 15, line 27, remove "with" in "handle with huge"*

*page 15, line 34, change "to further" to "for further"*

**Author's response.**

Thank you very much! These corrections have been applied to the text.

---

## Author Comment (AC2) · 26 Jun 2018

Maxim Kaurkin

**maksim.kaurkin@phystech.edu**

We thank the reviewer for a detailed analysis of the article and valuable comments. The paper has undergone a major reformulation and revision of English wording, some sections have been extended. Two figures and one section have been added, so the numbering has changed.

The following is the point-by-point response to the reviewers' comments (shown in "italic").

> *Referee comment.*
>
> *A.1) The description of CMF2.0, results of Test I and II and related conclusions need to be clarified:*
>
> *- First, I agree with Referee #1 that figure 1 is misleading as it suggests a 1:1 connection between the models processes and the coupler processes. I don't understand either the "which means data locality" on P.4, L.50, as there is certainly some exchange of data needed between the component processes and the coupler processes.*

**Author's response.**

Yes, we agree. The figure has been redrawn to show the basic idea that every coupler core communicates with a fixed subset of each component's cores. For example, on the Figure 1 the 1st core of the coupler (c1) sends and receives data from the component cores o1, o4, o7, i1, a1 and a4. Originally, we embedded this meaning in the term "locality", but indeed it can be misleading. So, the paragraph has been extended and reformulated in "master-slave" terminology without a reference to locality.

> *Referee comment.*
>
> *- P.5, L.63-66: these two paragraphs are not clear at all. What does "a subset of component's cores works only with individual master core in the coupler" mean? What does  n´ for the two cases, of the source and destination type  ż mean and why do you put a reference to Craig et al. 2005 here?*

**Author's response.**

These paragraphs have been reformulated. We are explaining the "master and its subset'" concept in the Section 3.2 (see previous comment), while these paragraphs are devoted to handling interpolation weights. The "source" and "destination" mapping types are briefly explained, while the thorough explanation can be found in (Craig et al., 2005, Section 3.4).

> *Referee comment.*

*- P.5., L. 69-70: You write "All necessary links are initialized at the beginning of run and are used at the calculation stage as persistent (Jacob et al., 2005). Âˈz I suppose that the links you mention here are not the SCRIP links? You should be using another word as this is confusing. Also, why do you mention Jacob et al. 2005 here?*

**Author's response.**

Yes, it is better to say "component-component SCRIP links and intracoupler rearrange routes". The paragraph has been reformulated. It refers to Section 3.3.2 of (Jacob et al. 2005), which can be accessed for a more detailed explanation of the rearranging concept.

**_Referee comment._**

*- P.5, L.72-73: I don't understand this sentence: "It is worth noting, that links are not sent directly, but as sorted unique cells vectors which allow one to avoid sending duplicated data. Âˈz. Again what "links" are you talking about here?*

**Author's response.**

It is just the data, which is being interpolated. The paragraph has been corrected.

**_Referee comment._**

*In Figure 2, 3 & 4 captions and on P.6, L.88, you should recall what is included in the timing. I suppose this is the time in seconds for the whole 10 model days for the whole ocean-atmosphere and atmosphere-ocean exchanges through the coupler. (Same remark for Figure 6).*

**Author's response.**

That is correct. Figures 2,3,4 and Figure 8 (previously 6) show the timing of 10-day model runs with disabled physics routines (see Section 3.3, 6th paragraph for the Test I description, and 9th paragraph for the description of Test II).

The corresponding remarks have been added to figure captions and to the mentioned place in the text.

**_Referee comment._**

*On p.6, L.92, you write: "It is clear that 20-40 coupler cores provide satisfactory speed for such problems, because ∼10 seconds costs . . ." It is not clear how you get these numbers. I see at best, i.e. with the MVS-10p_16, something between ∼15 and ∼20 seconds. Same remark for the number presented on P.7, L.4-5. This re-joins referee#1's comment about the fact that the log scale does not allow one to get the detailed information mentioned in the text.*

**Author's response.**

In our opinion, the logarithmic scale with time vs. number of cores is preferable for the analysis of parallel efficiency, because it allows simultaneously to compare with linear trend (which illustrates "perfect scaling") and to know the time of execution (in contrast to a speed-up plot or a nonlogarithmic time-cores plot). For convenience of estimations, we have added projection lines to the plots on the Figures 2 and 8. For example, on Figure 2, 32 cores of MVS-10p_16 under CMF2.0 give 10.5 seconds. The data of Figure 4 additionally has been presented in a table.

**Referee comment.**

*P.7, L.6: again here I don't understand why you write "perform only local communication"*

**Author's response.**

Since the master-slave algorithm was described above, this paragraph has gained a minor reformulation, which removes the reference to locality:

"Since every coupler core communicates only with a subset of

component cores, increasing of the coupler communicator size leads both to decomposing of the interpolation computations

and to decreasing of the component-coupler communication overhead, though slightly increasing intracoupler rearrangement

communications."

**Referee comment.**

*Section 3.4 on I/O should be completely revised. The long theoretical section on P.8 with detailed formula is not useful here, especially as you finally simply state "Asynchronous scheme was incorporated in the latest version our framework Âż without giving finally any results! The theoretical section should be cut and numbers obtained for the grid sizes you list for INMIO World ocean model should be provided.*

**Author's response.**

We have reduced and rewritten this section, so that it is now devoted to the CMF asynchronous scheme only. Data writing speed test results have been added for the CMF2.0 on the MVS-10P and BlueGene/P systems (new Figure 5). In Section 4.2 we have also added and discussed results of the actual asynchrony test of the CMF3.0 I/O system in conditions of the 0.1-degree global ocean model (new Figure 7).

**Referee comment.**

*The different utility modules available should be briefly described or a reference to a documentation or User Guide should be provided.*

**Author's response.**

We have provided the User Guide, which includes the description of the interface, utility modules and capabilities of the CMF3.0 system, as well as the process of installing and configuring the coupled model, as part of which the CMF is distributed. This manual is available at http://model.ocean.ru:6623/VITIM-manual-eng.pdf

*Referee comment.*

*A.2) The description of CMF3.0 needs to be extended*

*This paper is supposed to mainly describe CMF3.0, or at least this is what the title implies but very little is written about it. It looks like the author was in a hurry to finish the paper. More details should certainly be given on how the I/O service work (1st paragraph on P.11) and on the Data Assimilation part (currently only 2 lines, P.11, L8-9). These improved descriptions should be backed up with performance results (as is done for the interpolation).*

**Author's response.**

We have added a new section, which describes the mechanism and opportunities of working with global arrays in CMF3.0 by means of the class Communicator_GA (currently, Section 4.3). The I/O service description also has been extended. Its performance test has been presented in the Section 4.2.

As for Data Assimilation, it is a large area of our research, which requires a separate publication. We suppose that the graph of parallel efficiency for the DAS service (as an important result for this work) and references to our published papers about the data assimilation problem are sufficient.

*Referee comment.*

*Also the discussion of the interpolation results should be extended and detailed. How do the author get to 2-3 seconds per modelling day on 20-50 CPL cores? (This is mentioned also in the conclusions P.15, L.28.) Why is it expected that results would be worse than for CMF2.0 (and "worse" should not be used here because it implies that results for CMF2.0 are bad and that results for CMF3.0 are even worse)? Is it because of the shift from MPI to PGAS? Or because the tests were performed on a different platform?*

**Author's response.**

For convenience, we have added projection lines to the Figure 8, thus they show 20-30 seconds per 10 model days. The decline in performance is expected due to the overhead of using GA-library (as an intermediate send/recv data representation) and due to deprecated MPI_SEND_INIT procedures in the CMF3.0. This is a sacrifice for the compact code representation and for convenience of adding new features (like Data assimilation or Nesting technology). We have extended the Section 4 in order to clarify these features of the CMF3.0 system.

We have used "less strong" instead of "worse".

*Referee comment.*

*Also, in the conclusions, you write: "The key part of it, coupler, has a sufficiently small code size for such programs (about 5000 lines of code with unit tests) and is able to manage the main parallel problems of the coupled modeling - synchronization, regridding and I/O." I don't understand why you write that the coupler manages the I/O as this is not the case in CMF3.0.*

**Author's response.**

This phrase must refer to the CMF2.0. It has been corrected.

> ***Referee comment.***
>
> *A.3) The whole text needs reviewing by a native English speaker. The style and wording needs revision as some sentences are simply not understandable (at least by me),*
>
> *e.g.:*
>
> *- P.3, L.16: "Unquestionable advantage of non-coupler design is the absence of interference in the user code": why "non-coupler design"? Also, with OASIS, there is some interference in the user code but the objective is to minimize it.*

**Author's response.**

We mean that there is no coupling through a standalone coupler in OASIS3-MCT, so the user does not have to reorganize his code according to standard interfaces (e.g., as required for the cpl7 coupler). The phrase has been changed to "Unquestionable advantage of this non-coupler design is the minimization of interference in the user code, since there is no need to adapt it to interfaces required by a coupler."

> ***Referee comment.***
>
> *- P.5, L.76: "Several ping-pong tests were carried out for interpolation system using coupled ocean-atmosphere model."*

**Author's response.**

The phrase has been refined for explanation of ping-pong test conditions: "The performance rate of the CMF2.0 interpolation system was evaluated in several "ping-pong" tests, in which the coupler was maintaining component-component exchanges of the INMIO-SLAV ocean-atmosphere model with disabled solvers of physics equations (similarly to the ping-pong test of OASIS3 in (Valcke, 2013))".

> ***Referee comment.***
>
> *P.6, L.95-96: "work of the sequential algorithm is only possible with restriction that memory is allocated only for interpolation block, which is impossible in practice Âˈz: I am not sure what this sentence means exactly and why you write that it is impossible in practice while you do get some results on 1 core; do you mean it would be impossible with real models as the interpolation per se would require all the available memory?*

**Author's response.**

Yes, to perform this test we had to switch off the allocation of all physical model arrays, except those particularly involved in the test. So, in real numerical experiments the node memory (at least on considered supercomputers) will be insufficient for both physics equations solving and work of the 1-core coupler. Possibly it is better to say "unlikely" instead of "impossible". The phrase has been refined.

*Referee comment.*

*- P.13, L.63: "but we are more interested in scalability of the program on perspective sizes of computational resources Â˙z*

**Author's response.**

The phrase has been reformulated indicating the saturation of the decomposition algorithm:

"Obviously, at high core counts the parallel efficiency curve experiences "flattering". But assuming that the time step of the model is 5 min., the result of the experiment leads, e.g., to quite satisfactory five simulated years per wall-clock day (SYPD) rate achieved on 20000 cores of the BlueGene/Q supercomputer."

*Referee comment.*

*P. 13, L.71: "Time evolution of the sea-ice surface temperature is described in the same way as in prescribed ocean experiments.":*

**Author's response.**

The sentence has been removed.

*Referee comment.*

*P. 13, last paragraph: please rephrase, the current sentence with the "min." is too difficult to follow.*

**Author's response.**

The sentence has been rephrased and split in two sentences:

"Every

72 min., nine 2D-arrays were transferred from the atmosphere to the ocean (components of wind stress, short- and long-wave

radiation, fluxes of sensible and latent heat, precipitation, evaporation, air temperature at 2 m). Conversely, every 144 min. three

2D-arrays were transferred from the ocean to the atmosphere (upper gridbox temperature, temperature and concentration of sea ice)."

*Referee comment.*

*P.2, L25: "Coupling through shared file or sequential component is acceptable . . . Â˙z could be  n´ Coupling through shared file with components executing sequentially is acceptable . . . Â˙z*

**Author's response.**

We imply slightly different meaning. Changed to "Coupling through a shared file or through a

sequential hub is acceptable..."

> ***Referee comment.***
>
> *P.2, L33: ". . . and their representation in the interfaced style understandable . . ."*
> *could be " . . . and their adaptation to the interface understandable . . ."*

**Author's response.**

Changed to "This approach requires some reorganization of the components' code and its adaptation to the interfaces understandable by the driver..."

(since we are talking about a set of standard interfaces through which all model procedures should be called).

> ***Referee comment.***
>
> *- P.2, L41: "GFDL FMS (Balaji, 2012) system additionally suggests fully parallel data storage with file post processing at the end of the run Â˙z could be  nIn the GFDL FMS (Balaji, 2012) system, fully parallel data storage with file post processing at the end of the run is offered Â˙z*
>
> *- P.3, L.21: "According to proposals of Earth System Modeling conference, (Valcke et al., 2012), . . ." could be "According to the analysis of coupling technologies for Earth System Modeling by Valcke et al. 2012, . . ."*
>
> *- P.6, L86: "Performance is based on a standard Intel Fortran compiler" should be moved to the next paragraph and could be "On all supercomputers, the coupled system was compiled standard Intel Fortran compiler."*
>
> *- P.6, L.90: "Increasing number of coupler size"; should be "Increasing number of coupler processes" or "Increasing the size of the coupler communicator"*
>
> *- P. 11, L. 12: "Optimizations regarded to ignore repeated cell requests are preserved" could be "Optimization regarding repeated cells are preserved".*

**Author's response.**

These issues have been corrected according to the reviewer's suggestions.

> ***Referee comment.***
>
> *- P. 13, L.56: "Latest version of INMIOWOM model was fully integrated to CMF Â˙z could be  n´ CMF2.0 was fully integrated in the latest version of INMIOWOM  ż*

**Author's response.**

Changed to "The latest version of INMIO WOM is distibuted in an integrated package together with the CMF2.0 and 3.0, all necessary libraries and a standardized folder structure facilitating the

adding of new model components (including adapter files for the CICE sea-ice model)."

> ***Referee comment.***
>
> *- P.14, 3rd line: "Ice model was built into the ocean model , land model – into the atmosphere model" could be "The ice model is integrated in the ocean model and the land model in the atmosphere model"*

**Author's response.**

Changed to "The sea ice was simulated by the INMIO built-in ice thermodynamics model, while the land processes were incorporated into the SLAV atmosphere model."

> ***Referee comment.***
>
> *- P.14, L.6: "structurize" could be "structure"*

> ***Referee comment.***
>
> *B) Other comments:*
>
> *B.1) References:*
>
> *- P.2, L.6: The reference to OASIS3 should be Valcke 2013 (i.e. p18, L45)*
>
> *- P.2, L.36: The reference to OASIS3-MCT should be Craig et al 2017: A. Craig, S. Valcke, L. Coquart, 2017: Development and performance of a new version of the OASIS coupler, OASIS3-MCT_3.0, Geosci. Model Dev., 10, 3297-3308, https://doi.org/10.5194/gmd-10-3297-2017, 2017.*
>
> *- P.3, L.7 (as annotated in the manuscript): Reference to ESMF should be the more recent one: Theurich, G., Deluca, C., Campbell, T., Liu, F., Saint, K., Verten- stein, M., Chen, J., Oehmke, R., Doyle, J., Whitcomb, T., Wall- craft, A., Iredell, M., Black, T., Da Silva, A. M., Clune, T., Fer- raro, R., Li, P., Kelley, M., Aleinov, I., Balaji, V., Zadeh, N., Ja- cob, R., Kirtman, B., Giraldo, F., McCarren, D., Sandgathe, S., Peckham, S., and Dunlap IV, R.: The Earth System Prediction Suite: Toward a Coordinated U.S. Modeling Capability, B. Am. Meteor. Soc., 97, 1229–1247, https://doi.org/10.1175/BAMS-D- 14-00164.1, 2016.*

**Author's response.**

These issues have been corrected according to the reviewer's suggestions.

> ***Referee comment.***
>
> *B.2) P.2, L.35-36: OASIS3-MCT proposes coupled system not only as single executable, so this sentence is misleading; the important feature is that the coupling functions are not provided by a standalone coupler but by a coupling library linked to the component models. Please correct.*

**Author's response.**

The paragraph has been rewritten as:

"The coupled system can be launched as a single or multiple executable without a separate coupler whose functions in this case are provided by by a coupling library and performed in parallel on a core subset of each model component. Such a solution was proposed in OASIS3-MCT (Craig et al., 2017). A high-level driver controlling system sequencing is not required in this case.

> ### *Referee comment.*
>
> *B.3) P.2, L.44-50: This list mixes functionalities (1. and 2.) and characteristics (3. and 4.); please reorganise.*

**Author's response.**

Sentence reworked.

"Thus, we can point out the necessary features of modern coupling frameworks:"

> ### *Referee comment.*
>
> *B.4) P.3, L.11: It is not really fair to write that the computational costs of CESM coupler are quite significant 20%, as the CESM coupler does not only perform coupling and remapping but also performs the surface flux computation.*

**Author's response.**

The phrase has been changed to "Tests showed that computational costs of the CESM coupler (including coupling, remapping and surface flux computation) are quite significant 20%..."

> ### *Referee comment.*
>
> *B.5) P.3, L. 21-24: these 3 lines do not give an appropriate summary of the analysis provided by Valcke et al. 2012. Please correct.*

**Author's response.**
The paragraph has been rewritten as:
"According to proposals of the Earth System Modeling conference (Valcke et al., 2012), today there are several common aspects in coupling software development: an ability to communicate data between components, regrid data, and manage the time evolution of the model integration. There is a lot of custom parallel coupling mechanisms, with either single or multiple executable approache. We selected the approach with single executable because it can simplify the program flow and give additional opportunities for performance optimization. Besides, we used NetCDF for parallel I/O and SCRIP (Spherical Coordinate Remapping and Interpolation Package) (Jones, 1999) for regridding, as done in OASIS3 (Valcke, 2013)."

*Referee comment.*

*B.6) P.4, L.42-43 & P.9, L.47: Either provide details on what interceptor or Template methods are or don't mention them; one should not have to read the reference (Gamma et al., 1995) to understand the sentence.*

**Author's response.**

The reference to interceptor methods has been removed. Section 3.1 has got some reformulations to clarify the meaning of Template methods.

*Referee comment.*

*B.7) P.5, L77: References here are misleading as they seem to imply that "Test I condition" are explicitly defined in Valcke et al., 2012 or in Craig et al. 2012, while they are not.*

**Author's response.**

The references have been changed to (Valcke et al., 2013) where the ping-pong test is explicitly defined.

*Referee comment.*

*B.8) P.6, L.93: You should not use the word "failure" here as the test does not fail, it is just very slow.*

**Author's response.**

"Failure" changed to "ineffectiveness".

*Referee comment.*

*B.9) Using very technical coding terms along the text does not help understanding it (e.g. P.4, L.45: "Component class"; P.9, L.63:"resulted in class Communicator"; p.10, L.77-78:"since all services in CMF3.0 inherit base class Service it also allows one to easily add new ¡z ; P.10, L.80 : receives data using Communicator Âż ; P.12, L.25 :  n´ NormalEvent Âż or Ân  SyncVarEvent Âz : P.12, L.27 :  nGenerators realize abstract class EventGenerator, so new specific generator subclasses could be easily added ); I think it would be better to explain the concept that using these abstract terms.*

We have extended and reformulated the explanation of these methods. Some terms have been excluded (Component class,  EventGenerator). The others (class Communicator_GA,  class Service, NormalEvent, SyncVarEvent) are kept, since they are names of the objects described and are needed as references in the text.

*Referee comment.*

*B.10) P.12, L.26: Reference to Griffies et al. is not useful here.*

**Author's response.**

Instead of this reference we pay attention to importance of supporting experiments with prescribed forcing referenced to the real calendar, e.g. the Drakkar Forcing Set (Dussin et al., 2016)

> ***Referee comment.***
>
> *B.11) P.12, section 5.1: What is the resolution of the INMIO World Ocean model in these tests?*

**Author's response.**

It is 0.1 degrees (added to the last paragraph of the section and to the Fig. 9 caption)

> ***Referee comment.***
>
> *B.12) P.15, L.25-27: please specify if these numbers apply to CMF2.0 or CMF3.0. On line 27, change "CPL3.0" for "CMF3.0"!*

**Author's response.**

They apply to CMF2.0. These issues have been corrected.

> ***Referee comment.***
>
> *B.13) In general, I think the section 6 on Conclusions and future work could be fleshed out.*

**Author's response.**

In our opinion, it is necessary to summarize the results. We tried to shorten and concrete this section.

> ***Referee comment.***
>
> *B.14) I think the "Code availability" section is not satisfactory regarding GMD standard but I will let the Topical Editor decide on this point.*

**Author's response.**

We can change this section if it is required.

> ***Referee comment.***
>
> *C) Minor comments:*
>
> *C.1) In the abstract, you write "As addition a parallel realisation of the EnOI (Ensemble Optimal Interpolation) data assimilation method as program service of CMF3.0 is presented." but this is not the only example presented in section 5.*

**Author's response.**

Corrected.

"As an addition, some information about the parallel realization of the EnOI (Ensemble Optimal Interpolation) data assimilation method and the nesting technology, as program services of the CMF3.0, are presented."

> ***Referee comment.***
>
> *C.2) P.3, L.13: It is not right to write that OASIS3 is the most popular version of OASIS as most groups are using OASIS3-MCT today.*

**Author's response.**

The sentence has been changed to "The OASIS3 system was very successful and was widely used by many research groups around the world."

> ***Referee comment.***
>
> *C.3) P.3, L.16: In OASIS3-MCT, MCT procedures are executed on all component model cores*

**Author's response.**

Corrected:

"The new version, OASIS3-MCT (Craig et al., 2017) resolves the issue of sequential interpolation by using MCT procedures executed on all model component cores, instead of mapping through a standalone coupler."

> ***Referee comment.***
>
> *C.4) P.4, L.33: Define SOA the first time it appears in the text*

**Author's response.**

Done (in the last paragraph of Section 2)

> ***Referee comment.***
>
> *C.5) P.2, L53: Define WOM the first time it appears in the text.*

**Author's response.**Done (in the beginning of Section 2)

---

## Author Comment (AC3) · 26 Jun 2018

**Vortex-Investigating Terra Integrated Model**
**VITIM 3.1.1**
**user guide**

K.V. Ushakov, V.V. Kalmykov, R.A. Ibrayev, and M.N. Kaurkin.

May 28, 2018

**Contents**

**1  Introduction**

This manual is a pilot version of the instruction for the user of the coupled geoscientific model VITIM. By studying it, you can sequentially go through all levels of the deployment of the model and immersion in the work, from installing the operating system to working with the internal content of the component modules and the coupler. At the end of initial levels, simple tests are provided in order to make sure, in the first approximation, that everything functions properly. All paths to folders and files in this manual will be listed from the folder **vitim3.1**, or files and folders will be given without paths.

**1.1  Some features of this release**

— Supported configurations: global20, WOM025_ice, WOM05_ice, arctic0125a, arctic025t, laptev0125(a,c) и WOM01.

— Supported work in the CMF2.0 and CMF3.0 frameworks.

— Sea ice models supported: Schrum and CICE.

— The CICE ice model may be launched on the reduced grid (relatively to the ocean one) for processor resources saving (only under CMF2.0).

— SCRIP interpolations can take into account the input and output land-sea masks.

— Diagnostic utilities o_diagn include: time averaging and flux calculations for meridional heat transport studies.

**2  User workplace preparation**

**2.1  Installing the operating system and compiler**

The subtleties of the operating system installation are beyond the scope of this guide, completely lying on your shoulders and communication channels with the Internet reference resources. The current version of VITIM installed on PC works with the Ubuntu 14.04 operating system. With newer releases of Ubuntu, there were difficulties, which, however, are likely to be overcome by a couple of lines in .bashrc. Therefore, the desire to deal with them on your part is warmly welcome. The model is configured for the Intel Fortran compiler. To work with CMF3.0, you will need version 15 or higher.

**2.2  SSH configuring**

SSH is a network protocol that allows you to work remotely with a certain system, for example, with a cluster. To connect to a remote machine, for example, we type:

```
ssh ivanov@mvs10p.jscc.ru
```

How can the system understand that you are really ivanov? You will either be asked to enter the password (issued to you by the administrator of the machine), or the system will ask for your private key. It is believed that the method with the key is safer. What is its essence? You generate a pair of keys with `ssh-keygen` command: public and private. The public one must be sent to the administrator of the machine, and the private one you keep in a safe place. When connecting, you show the private key. If the pair has matched, you enter the cluster. When connecting, an error of the following kind may appear: WARNING: UNPROTECTED PRIVATE KEY FILE! This means that ssh has detected that your data in the ∼/**.ssh** folder has too open access rights and can be read directly without any hacking. Therefore, you have to set the right permissions on the folder and its contents (read about access rights)

```
chmod 700 ~/.ssh && chmod 600 ~/.ssh/*
```

How can we simplify the work with ssh? For each host, we create an entry in the configuration file. Whenever possible, ssh will access the information in this config file, minimizing the number of user-entered parameters. For example, if you need to present a private key to log in, then you add the following lines to the file ∼/.ssh/**config**:

```
Host inmio
    HostName 83.149.207.46
    Port 22
    User ivanov
    IdentityFile ~/.ssh/ivanov_inmio_key
```

This is what you need to do in order to gain access to the GROSM server, which stores the model codes, installation scripts, atmospheric forcing data, etc. Generate a key pair and ask the administrator (sherema@yandex.ru) to register the public key.

Let's check:

```
angarsk@angarsk:~$ ssh inmio
Welcome to Ubuntu 16.04 LTS (GNU/Linux 4.4.0-21-generic x86_64)

 * Documentation:  https://help.ubuntu.com/

739 packages can be updated.
384 updates are security updates.

Last login: Fri Mar  2 15:26:27 2018 from 91.225.112.
```

**2.3   Installing geophysical software**

The model uses a number of libraries for I/O, interprocessor communications, data processing, etc. For convenience, a script is provided, which downloads them, installs and sets environment variables. Download the latest version of the script and ancillary files from the remote repository on the GrOSM server:

```
git clone ssh://inmio/git/scripts.git
```

To configure the installation script, it is sufficient to set several main parameters (for example, the necessary versions of the compilers: mpif90 or mpiifort, etc.) in the script entries. At this stage, you must have the necessary compilers installed. On a PC, it is the Intel Complier, and on the cluster − loaded modules, usually specified for convenience in the .bashrc file. Now you can run the installation script. In this example, the software will be installed into the folder **$HOME/House/software**

```
bash install_soft.sh --install-dir $HOME/software --all \
--mpifc [Fortran compiler] --mpicc [C compiler] --mpicxx [C++ compiler]
```

Usually, for Intel compilers, the following parameters are suitable:
[Fortran compiler] = mpiifort
[C compiler] = mpiicc
[C++ compiler] = mpiicpc

By default, the script will try to download data from remote servers. If there is no Internet connection, manually transfer the library archives to the cluster and try again. Where to get the archives? Run on an Internet-connected system the script with parameters `--all --only-download`. The required archives will appear in the target installation folder. At the end of the installation, tests are automatically performed. Make sure they are completed without errors:

```
Testing NetCDF...SUCCESS
+ mpiifort -c -cpp test_ga.f90 -I/home/angarsk/House/software/include
+ mpiifort -o test_ga.exe test_ga.o -L/home/angarsk/House/software/lib
+ mpiexec.hydra -np 4 ./test_ga.exe
 Testing GlobalArrays...GA_INIT
 Testing GlobalArrays...MA_INIT
 Testing GlobalArrays...NGA_CREATE_IRREG
 Testing GlobalArrays...NGA_DISTR
 Testing GlobalArrays...GA_PUT
 Testing GlobalArrays...GA_SPD_INVERT            0
 Testing GlobalArrays...NGA_GATHER
SUCCESS
+ rm -f simple_xy_par.nc test_ga.exe test_netcdf.exe '*.mod' test_ga.o tes
+ set +x
===============
ALL TESTS PASSED
===============
```

As a result, the software and auxiliary scripts are installed in the **$HOME/House/software** folder. Close the console and reopen it, so that the necessary paths are registered on your system. Now, typing, for example, `which cdo`, you'll find that the path leads to your software folder. Let's check:

```
angarsk@angarsk:~$ which cdo
/home/angarsk/House/software/bin/cdo
angarsk@angarsk:~$ which h5dump
/home/angarsk/House/software/bin/h5dump
angarsk@angarsk:~$ which ncdump
/home/angarsk/House/software/bin/ncdump
angarsk@angarsk:~$ locate job_launcher.sh
/home/angarsk/House/software/bin/job_launcher.sh
/home/angarsk/Plots/scripts/launchers/job_launcher.sh
```

**2.4   Downloading the model and the geophysical data**

The user workstation is the folder **vitim3.1**, in which all models and data are stored. To get it, download the workpiece from the repository:

```
git clone inmio:/git/vitim3.1
```

Download the latest versions of models from the repositories to the folder **comps**:

```
cd vitim3.1/comps/ocn
git clone inmio:/git/inmio4.1.git
cd ../ice
git clone -b vitim2.1 inmio:/git/cice-5.1.git
```

For the minimum run, the topography ETOPO, the Levitus data WOA2009 and the CNYFv2 forcing should be in the **vitim3.1/data_external** folder. If this data already exists on your computer (cluster), then put in the **data_external** symbolic links to its locations. If not, run the download scripts (unpacking the forcing will take a few minutes):

```
cd ../../data_external
bash get_IC_databases.sh
bash get_forcing_databases.sh
cd ../coupling
```

In the future, when working with more advanced model configurations, make sure that the **data_external** folder has the necessary forcing or links to it.

**3   How to choose the model configuration and start working**

The model comes with a set of several standard configurations with different computational domains, resolution, forcing and enabled parameterizations. When you first get acquainted, you just need to choose one of them. Pay attention, however, to the resolution: grids with a size greater than **200 × 100** usually do not fit into the RAM on the PC.

In the file **coupling/config**, specify the full path to the **vitim3.1** folder and select the configuration of the numerical experiment (uncomment the corresponding line). For example, the Laptev Sea with a grid size of **40 × 60**:

```
export VITIM_PATH=~/VITIM3.1/vitim3.1
...
export RES="laptev0125c"; export GRID="40x60"; export INMIOCOUPLED="yes"
```

Now create symbolic links among model files for this configuration.

```
bash links_inmio set
```

Your further steps depend on the version of the Compact Modeling Framework (CMF).

**3.1   Compiling and running under CMF2.0**

Compile the stand-alone ocean model (i.e. coupled model with non-interactive atmosphere and land runoff components):

```
cd coupling
./makeclean_all
```

A complete set of compilation commands will be executed with preliminary removal of the object files. In the future, you can call **./make_all** – the system will recompile only changed files (see, however, the note in the section 4.1). Make sure to see the message "Coupled model for component set <ocn atm_ncar lnd_core ice_cice> compiled successfully."

Now create the files of initial conditions and interpolation weights. Warning: when this command is executed, all nc-files in the folders (symlinks) **coupling/data** and **coupling/off/data** will be deleted.

```
cd configure
./generate_laptev0125c
```

In this example, the script for generating grids for the laptev0125c configuration is called. For other configurations, scripts are named in a similar way and stored in their instances of the **configure** folder (links to which, as you already know, are activated when you call `bash links_inmio set`). If there occurs a library error, try to call `make clean` in the **off/SCRIP/source** folder. Check that there are now 5 files with weights, the initial conditions, and an active link to forcing in the **coupling/data** folder:

```
angarsk@angarsk:~/VITIM3.1/vitim3.1/coupling/data$ ls -1
ATM_NCAR_192x94_to_ICE_CICE_40x60.nc
ATM_NCAR_192x94_to_OCN_40x60.nc
CNYFv2
ICE_CICE_40x60_to_OCN_40x60.nc
LND_CORE_360x180_to_OCN_40x60.nc
OCN_40x60_to_ICE_CICE_40x60.nc
OCN_40x60x49_IC.nc
```

Create symbolic links for CICE ice model. Warning: if this step is forgotten, the model may start without visible troubles, but produce incorrect results!

```
bash links_cice set
```

After this step, there must not be any broken symbolic links in the **coupling** folder. The model launch is performed from the folder **coupling**. This is an example of the launch command:

```
mpirun -np 7 ./model.exe CPL 1 OCN 2 ATM 1 LND 1 ICE 2 abc 0 0 3
```

Here abc is the name of the numerical experiment (arbitrary 3 symbols), 0 0 3 – experiment duration (model years, months and days).

**3.2   Compiling and running under CMF3.0**

Compile CICE and the coupled ocean-ice model with non-interactive atmosphere and runoff components:

```
cd coupling/comp_ice_cice/cice
./comp_ice
cd ../..
bash make ocn atm_ncar lnd_core ice_cice --clean
```

Make sure that the message appeared: `"Executable <.//cpl.exe> was created successfully."`
Now create the files of initial conditions and interpolation weights. Warning: when this command is executed, all nc-files in the folders (symlinks) **coupling/data** и **coupling/off/data** will be deleted.

```
cd off
bash generate_ocean_all.sh 40 60 49 abc
```

In this example, 40 and 60 are the horizontal grid sizes (should be the same as in the configuration you selected in the **coupling/config** file), 49 - the number of vertical levels (so far only this option is available), abc – the name of your experiment (any combination of 3 numbers or letters). If there occurs a library error, try to call `make clean` in the **off/SCRIP/source** folder. Check that there are now 5 files with weights, initial conditions, and also an active link to the forcing in **coupling/data**:

```
angarsk@angarsk:~/VITIM3.1/vitim3.1/coupling/data$ ls -1
ATM_NCAR_192x94_to_ICE_CICE_40x60.nc
ATM_NCAR_192x94_to_OCN_40x60.nc
CNYFv2
ICE_CICE_40x60_to_OCN_40x60.nc
LND_CORE_360x180_to_OCN_40x60.nc
OCN_40x60_to_ICE_CICE_40x60.nc
OCN_abc_IC.nc
```

Create symbolic links for CICE ice model. Warning: if this step is forgotten, the model may start without visible troubles, but produce incorrect results!

```
bash links_cice set
```

After this step, there must not be any broken symbolic links in the **coupling** folder. Into the folder **coupling/configure** it is necessary to put a namelist file for the given name of the experiment (in our example it is **exp_abc.in**). Its contents are easily readable, examples are given in the **coupling/configure** folders for all basic configurations.

The launch is performed from the **coupling** folder. Here is an example of the launch command:

```
bash job_launcher.sh --machine ubu --np 9 --exe ./cpl.exe
    --args "DTR 1 CPL 1 IOD 1 OCN 2 ATM 1 LND 1 ICE 2 abc"
```

**4 Basic options**

**4.1 Launching on various numbers of tasks**

In the example above, you can specify other cores (tasks) numbers for OCN, ATM, LND, ICE components. The utility **coupling/comp_cpl/bin/test_decomp.exe**, which is created when the system is compiled under CMF2.0, will tell you the valid number of cores and the corresponding subdomain sizes. For CICE operation, it is necessary that the number of cores and subdomain sizes of one core of the ice model be specified in the file **coupling/comp_ice_cice/cice/comp_ice** for the selected configuration, and the number of cores also specified in the
**coupling/comp_ice_cice/cice/input_templates/*name_of_configuration*/ice_in**.
  After every such reconfiguring (affecting CICE), you need to rebuild completely the CICE and the coupled model. In the command line parameter `np` at launch, do not forget to specify the total number of cores for the coupled model. If you do not need to run one of the components, its name and the number of cores in the launch command are not listed. In particular, if you do not specify `ICE`, the ocean built-in thermodynamic ice model of C. Schrum will work.

**4.2 Selecting atmospheric and runoff forcing**

In the 2.4 section, we have downloaded the atmospheric and river data of CNYFv2 – the normal CORE-I year. The standard model configuration can also work with IAFv2 – "real" synoptic data of the CORE-II protocol, based on reanalysis and observations for 1948-2009
(http://data1.gfdl.noaa.gov/nomads/forms/core/COREv2/CIAF_v2.html). They occupy about 30 GB and, usually, are available on clusters used by the GrOSM. To select the database that your `atm_ncar` and `lnd_core` models will read, specify the parameters of the named lists `atm_forcing_type` and `lnd_rivers_type` in the files **coupling/configure/atm_list.in** and **coupling/configure/lnd_list.in**, respectively. The description of the main parameters of the named lists of the model is given in the appendix A.

**4.3 INMIO built-in ice model**

If you do not specify the `ICE` and its number of cores in the launch command, the ocean built-in thermodynamic ice model by C. Shrum will work. In this case, to save memory when generating grids (especially for the WOM01 configuration), you can disable the generation of CICE grids: switch `ice_grid` to `.false.` in **comps/ocn/inmio4.1/driver_cmf2.0/off_ocn_module.f90** (in the CMF2.0 environment) or **comps/ocn/inmio4.1/driver_cmf3.0/cmf_ocn_off_adapter.f90** (in the CMF3.0 environment). If you work with CICE, make sure that `ice_grid = .true.`

**4.4 Working on reduced ice grid (available under CMF2.0 only)**

Several standard configurations, for saving resources, by default work on the reduced ice grid of CICE, which covers only the northern polar cap. In the rest of the calculation domain, in this case, the built-in ice model by C. Schrum works. When you turn on a standard reduced-grid configuration

— Check the flag `cice_and_schrum` in
**comps/ocn/inmio4.1/driver_cmf2.0/o_par_module.f90**. For a reduced grid it must be `.true.`, for a regular grid `.false.`

— Check the value of `additional_ny` at the top of the
**coupling/comp_ice_cice/off_ice_cice_module.f90** file. For a regular grid it must be 0, for a reduced one – see recommended values in comments.

— To work on a reduced CICE grid, it is necessary that the decomposition of the ocean with respect to the j axis consists of at least two bands (in order for one to hold the ice of CICE and the other to work with Schrum model). Check this with the **test_decomp.exe** utility.

If you want to *enable* grid reduction for a particular configuration (for example, take a global regular model and use it to explore the Arctic), then do the following. It is recommended that you change these settings carefully only if you understand what is happening. To enable reduction:

— In the file **off_ice_cice_module.f90** specify `additional_ny` – a negative integer, meaning how many rows must be removed from the southern side of the grid

— Specify the new j-size of the grid in **config** and in **ice_list.in**, new block sizes in **comp_ice**.

— Add commands for changing CICE sizes in the names of three interpolation files at the end of the grid generation script(see examples in the `global20` and `WOMO25_ice` configurations)

To turn reduction off, take everything back: set `additional_ny=0`, the ice grid size equal to the ocean one, interpolation files are not renamed.

**4.5   Working in offline analysis mode (available under CMF2.0 only)**

If you run the model in offline analysis mode, then the value of the key in the file **coupling/analysis_flag** should be equal to `.false.` If in the normal calculation mode, then `.true.`

**5   Compact Modelling Framework CMF3.0**

The Compact Modelling Framework performs two main tasks:

— Support of service operations for a particular model (for example, working with the file system). The CMF allows you to clearly separate the code of physical model (for example, the ocean) and the code responsible for the technology (for example, the procedure for saving data). This separation, firstly, simplifies the architecture (each module deals with its own business), and, secondly, gives the possibility for the developer of service modules to modify their insides without interfering with the physical model.

— Support for coupling of models (for example, creating an ocean-atmosphere model). Historically, models are separate programs that calculate their own physics (ocean model models the ocean). How to join two independent models so that they work together? One approach is to connect them using an adapter (similar to an electrical adapter) to a modelling framework. As a result, within itself each model continues to consider its physics, and through the adapter it communicates with the other participants of the coupled model.

**5.1   Getting started with the CMF3.0**

To understand how the system works and what is needed to run it, it's better to use specific examples. Example 1 shows a sequence of actions that allows you to run a simple CMF test from scratch. The following examples explain how you can complicate this workflow to take full advantage of the CMF capabilities. The last example shows how to connect your model to the system.

**Example 1: running an empty atmosphere-ocean coupled model**

In this example, we show how to run from scratch a simple CMF3.0 test, simulating the launch of two models (ocean and atmosphere) that do nothing. Go to the model folder and run:

```
cd test_suit
bash tester.sh --t empty_comps --clean --test
```

What happens in this test? A special script is launched, which allows to combine the model build and its launch (this script is made for convenience, now it is not necessary to deep into how it works). The script goes into the folder `empty_comps`, containing a test (but from the point of view of the system quite full) version of the coupled ocean-atmosphere model. In the folder there is a simple script describing the test (`test_description.sh`), which looks like this:

```
COMPS_BUILD="ocn_test atm_ncar"
RUN_COMMAND[1]="-np 5 ./cpl.exe DTR 1 IOD 1 CPL 1 OCN 1 ATM 1 tst"
```

That is, the script **tester.sh** sees that it has to build a coupled model from the **ocn_test** and **atm_ncar** folders and run it on 5 cores, giving services (not yet think about this), ocean and atmosphere 1 core to each. In the console, you will see that the system has output the experiment parameters and entered the computational cycle phase, which in this test consists only of receiving STOP signals (normal termination) from the models. Result: You just started the *hello-world* example of a coupled model. The model did nothing, but only sent a signal about the normal completion. In the following examples we will add work for it.

**Example 2: teaching the model to save diagnostics**

In the previous example, the model was simply connected to the CMF using an adapter, ran off the assigned time of the experiment and ended. Where are these actions described? The logic of any model is described with the help of a special adapter class, which, as befits an adapter, knows how to connect to the system and at the same time has inputs for the physical model. To understand the further process, it is helpful to read the first sections of the manual about the model component.

Now you roughly understand the logic of the system and it's time to look at the code of the previous example. Open the file **/empty_comps/comp_ocn_test/cmf_ocn_test_cpl_adapter.f90**. You see the implementation of the very interfaces that are described in the manual. In this example, they are empty (hence the name of the test). The only non-empty method calls `ini_reg_comp` to register the model in the system. It is thanks to this registration that the system knows how many steps requested to run this empty model and what are sizes of its arrays.

Now, let's teach the model to write the diagnostics and for this we will slightly complicate the code. To do this, create a new test (already created for you) and call it **/save_dg**. The logic of the new model is not much more complicated and is described in the file **/empty_comps/comp_ocn_test/cmf_ocn_test_cpl_adapter.f90**. In addition to registering the model, we added 2 arrays (2D and 3D), code that allocates memory for them, and added registration of events over these arrays. Here appears the second important property of CMF, namely the ability to say "I want this array to be written every 2 hours to a file". To understand what this is about, be sure to read the beginning of the section on system events. Now you can go on to what the example does. In the implementation of the interface `ini_reg_data` for a 2D array we see:

```
call this % register_array(arr_name = "test_dg_2D", indexing = "ij",
        arr = save_dg_2D(iwest:ieast, jsouth:jnorth))
call this % register_periodic_event(arr_name = "test_dg_2D", act = "SAVE_DG", dh = 1)
```

The first line registers the array `save_dg_2D` under the name "`test_dg_2D`" in the system (in fact, the system remembers its address and indexing). The second line registers a periodic event over this array, saying that every 1 hour you need to create a diagnostic save event (namely, take the array at the address, send it to the appropriate service for saving to the diagnostic file). You can run this test, but first look at the file **test_description.sh**:

```
COMPS_BUILD="ocn_test"
RUN_COMMAND[1]="-np 4 ./cpl.exe DTR 1 OCN 1 IOD 1 CPL 1 tst"
....
```

We ask to build only the ocean model and run it on 1 core (the remaining cores are service ones). Below, the file describes the conditions checking results (namely, that for different numbers of cores we get the same diagnostic file). Now you do not have to think about it. Running:

```
cd test_suit
bash tester.sh --t save_dg --clean --test
```

The script will run 5 different tests, comparing the results with the first one, and will report the results. The result: we figured out how the system understands where is the model and how to work with it, learned how to generate events and launched the first adequate model that flushes data to disk.

**Example \*: connecting your model**

Until now, we have run built-in tests. Now you can create your own model (as the main model at the root of the system, or while in the same form of a test). To do this:

**Create a backbone of the derived class**
To create a new model, a script is provided (it is available if you installed the software correctly):

```
bash generate_comp.sh ocn_test
```

The script creates the model folder, all subfolders needed, two derived class templates (for the component and the off-block) and **makefile** building it all. You can immediately execute `make` – an ocean model library will be compiled that does nothing and consists of two files. Actually, all the models from the previous examples began with the call of this script.

**Fill in the derived class**
Fill in every class method according to your model requirements.

**Connect a component to the system**
The compilation system is constructed in such a way that if you add your model (for example, the ocean), then at compilation it is sufficient to specify the name of the folder, which has an agreed form. During compiling, the build script takes the name of your version of the component and understands that the first three letters denote the component of the Earth system (`ocn, atm, ice, lnd`), and then your version (`ncar, test, inmio`). The script will go into the desired folder (for example, **comp_ocn_test**), build the library there, and tell the main program that the compilation will involve the ocean model, and its version (for example, `test`). All these conventions are automatically applied if you call **generate_comp.sh**. The main script of the compilation system **make** takes the names of the folders (models). Actually, it is called by the script **tester.sh**, which used in examples above. For example, if we created a new ocean model `ocn_test` at the root of the system, we can build and run in this way:

```
bash make ocn_test
mpirun -np 4 ./exe OCN 1 CPL 1 DTR 1 IOD 1 exp
```

When calling the ocean procedures, will be used exactly the version of the ocean that was transferred to the compiling script (namely, `ocn_test`).

**5.2   Model components**

**General idea**

How can CMF learn about your model (for example, the ocean) and, accordingly, help it to perform service activities and communicate with other models?

One approach is to define a special adapter class, or in other words, a generic model (component). Such a component is a model skeleton and defines only its interface, but not implementation. The system will be able to call the adapter class methods (because it knows the interface), which are very general actions, for example, performing the entire initialization or one full physics step. At the same time, the system does not know what exactly will be done inside these methods – it leaves their implementation at the discretion of the model (for example, the calculation of thermodynamics and dynamics in the main step).

In practice, the described process becomes an inheritance of the class `Component` and implementation of abstract interfaces. Each model defines them at its own discretion. Plus, the user can call convenient methods defined for him in the `Component` class (for example, registering an event). As a result, to work in the system it is enough to create one adapter class that calls the specific methods of the model (your physical procedures) and the auxiliary methods of the system (defined for convenience in the class `Component`). The logic of the class will determine the logic of the work of your model in the CMF.

**Component class interface**

Below are some of the interfaces of the Component class:

```
! Abstract methods that must be implemented in the model

! Abstract method for registering the model in the system
! must call register_model()
procedure(I_ini_reg_comp), DEFERRED :: ini_reg_comp

! The abstract method for executing all allocate()'s in the model,
! since further it will be necessary to transfer addresses
procedure(I_ini_allocate), DEFERRED :: ini_allocate

! Abstract method for registering all data and mapping events in the system
! must call register_array(), register_event()
procedure(I_ini_reg_data), DEFERRED :: ini_reg_data

! Abstract method for all user-defined initializations
procedure(I_ini_main), DEFERRED :: ini_main

! Abstract method for one physical model step
procedure(I_make_step), DEFERRED :: make_step

! Abstract method for all finalizing actions
procedure(I_finalize), DEFERRED :: finalize

! Auxiliary methods of the base class that can be called from the model

! Registers a model in the system
procedure, public :: register_model

! Registers an array in the system
generic, public :: register_array => ...

! Creates a generator for time-uniform events
procedure, public :: register_periodic_event

! Creates a generator to bind to the time axis of a netCDF file
procedure, public :: register_synced_event

! Generates a single event
procedure, public :: raise_event
```

**More on auxiliary methods**

```
procedure, public :: register_model

 Description:        Registers a model in the system
 Arguments:
  *_size          - model array sizes
  decomp_type     - decomposition type ("1D" or "2D")
  timestep_sec    - time step, seconds
```

```
generic, public :: register_array

 Description:        Registers an array in the system, saving its parameters
                     (address, attributes) under tag <arr_name>
 Arguments:
  arr_name          - array name (string name, not Fortran name)
  indexing          - indexing ("ijk" or "kij")
  arr               - the Fortran array itself (its address)
```

```
generic, public :: register_periodic_event

 Description:        Creates a generator for time-uniform events
 Arguments:
  arr_name          - string name of a registered array
  act               - the action to be done at the moment of the event,
                      e.g, ''SAVE_DG''
  src               - (optional) data source (file, other component),
                      e.g., ''/data/ocn_test_data.nc''
  dst               - (optional) data receiver
  info              - (optional) any other information
  dh, dm, ds        - (optional) event period (hours, or minutes, or seconds. If all
                      equal to 0, the the event will occur only once in the beginning
                      of the run (e.g., ''READ_CP'').
```

```
procedure, public :: register_synced_event

 Description:        Creates a generator for event that is bind to the
                     time axis of a netCDF-file
 Arguments:
  arr_name          - string name of a registered array
  src               - (optional) file data source,
                      e.g., ''/data/ncar_temp.nc''
  start_date        - (optional) from which date to start binding.
                      By default, from the start of experiment.
```

```
procedure, public :: raise_event

 Description:        Raises a user-defined event.
 Arguments:
  arr_name          - string name of a registered array
  src               - (optional) data source (file, other component),
                      e.g., ''/data/ocn_test_data.nc''
  dst               - (optional) data receiver
  dt_rec            - (optional) date record (if we want to take file data related to
                      a specific date)
```

**For system developers**

**How the time cycle of the model looks**

After all initializations, the component enters the main time cycle `model_cycle`:

```
subroutine model_cycle(this)
    ! Parameters skipped for brevity

    ! Sending all arrays for registration to services
    call this % ev_scheduler_ % get_all_events(new_events)
    do i = 1, new_events % length()
        ev = new_events % get(i)
        call this % send_request(ev)
        call this % try_register_comp_ga(ev)
    end do
    call this % raise_event(act = "STOP")
    call CompSplitter % barrier()

    do while (.TRUE.)

        ! Ask Scheduler to collect all events of the current step
        call this % ev_scheduler_ % gather_events(this % model_time(), new_events)

        ! Process events on the side of the model
        do i = 1, new_events % length()
            call this % handle_event(new_events % get(i))
        end do

        ! While timer @model_clock_@ ticks, we are working in the cycle
        if (this % model_clock_ % is_stopped()) EXIT

        ! Calling physical model timestep
        call this % make_step()

        ! Clock ticks
        call this % model_clock_ % tick()
    end do

    ! At the end of the cycle, we notify the services
    ! that the model has completed its work normally
    call this % raise_event(act = "STOP")

end subroutine
```

**How the model reacts to events**

As described in the Events section, after an event is generated, it is first processed by the model. To do this, the `Component` class defines the method `handle_event()` (short code):

```
subroutine handle_event(this, ev)
    ...

    ! Different types of events lead to different reaction
    select case (ev % action())

        ! Sending events: wait until the ga-array is freed, put the data there,
        ! synchronize and mark the array as full,
        ! send request to the corresponding service
        case("SAVE_CP", "SAVE_DG", "SEND_MP")
```

```fortran
            do while (TRIM(this % comm_ % get_info(ev % ga_name(), &
                COMM_GA_STATUS)) /= "free"); end do
            call this % put_to_ga(ev)
            call this % comm_ % sync(CompSplitter % i_am_id())
            if (CompSplitter % is_first_rank()) &
                call this % comm_ % put_info(ev % ga_name(), COMM_GA_STATUS, "full")
            call this % send_request(ev)

        ! Receiving events: send the request to the service, wait until the ga-array
        ! is full, get data from it, synchronize and mark the array as free
        case("READ_FD")
            call this % send_request(ev)

            do while (TRIM(this % comm_ % get_info(ev % ga_name(), &
                COMM_GA_STATUS)) /= "full"); end do
            call this % get_from_ga(ev)
            call this % comm_ % sync(CompSplitter % i_am_id())
            if(CompSplitter % is_first_rank()) &
                call this % comm_ % put_info(ev % ga_name(), COMM_GA_STATUS, "free")

        ! Receiving of mapping: there is no request, we simply wait until the ga-array is full,
        ! get data from it, synchronize and mark the array as free
        case("RECV_MP")
            ! This is push-event, so no request:
            ! just register, wait, get, mark as free

            do while (TRIM(this % comm_ % get_info(ev % ga_name(), &
                COMM_GA_STATUS)) == "free")
                ! call Debugger % log_msg("Current GA status is: &
                ! "//TRIM(this % comm_ % get_info(ev % ga_name(),&
                ! COMM_GA_STATUS)))
            end do
            call this % get_from_ga(ev)

            call this % comm_ % sync(CompSplitter % i_am_id())
            if(CompSplitter % is_first_rank()) &
                call this % comm_ % put_info(ev % ga_name(), COMM_GA_STATUS, "free")

        ! Just send request and exit
        case("STOP")
            call this % send_request(ev)

        ! Send the request and wait for the service to finish,
        ! because we can not continue working
        case("ERROR")
            call this % send_request(ev)
            call CompSplitter % barrier()

        case default
            call this % fatal_error("raise event: &
                uknown action: <"//TRIM(ev % action())//">")
    end select
end subroutine
```

**How to change the reaction of the model to events**

Warning: In this paragraph, changes are made to the system code. If you are not sure about your actions, contact the developer. To change the response or add a new behavior, simply expand the `handle_event()` method.

Warning: mind the synchronization issues! The model should be blocked if the ga-array (which is the exchange buffer) is still occupied (for the "put data" event) and, conversely, is free (for the "get data" event). For this, the status of the information array is checked. If you do not set a lock, there is no guarantee that the model will receive or send complete data. After the model has put or get the data, it must change the status of the ga-array accordingly.

**Note: adding a new component of the Earth system**

Warning: In this paragraph, changes are made to the system code. If you are not sure about your actions, contact the developer. In the file `ComponentSplitter` add the required component name:

```
character(3), parameter :: COMPONENT_NAMES(9) = &
    (/ "DTR", "CPL", "IOD", "OCN", "ATM", "ICE", "LND", "SEA", "TST"/)
```

**Notes for the system developer**

If in the future there will be an opportunity to get rid of explicit synchronization through an array of information, it will be good. Now this approach is chosen, since we can not "lose data". That is, even if some component (ocean model) is faster than another component (atmosphere model or IOD, which slowly writes data to a file), we do not have the right to lose the array. The accumulation of arrays in the form of a queue will also lead to nothing, since models usually work at a constant speed and as a result, the queue will simply exhaust all available memory. Therefore now, if the "fast" model is ready to put data, but the ga-buffer is still occupied, it is blocked.

**5.3 Events in the system**

**General idea**

Events are messages about the need to perform certain actions on an array of data. For example, when we want to send model data for saving to a diagnostic file, we generate (raise) an event with the type `SAVE_DG` and some parameters (for example, a destination file). Events are produced on the side of the model: by an unpredictable call of the `raise_event()` from the user (for example, when a critical drop in the level in the ocean occurs), or by a generator (but, in fact, by the same `raise_event()`. Note: both the model and the services have their own reaction to events (see the sections about the Model Component and Services).

The first to respond to the event is the model. It looks at its type and determines what to do (e.g.,in case of `SAVE_DG` – put data into the ga-array, send a request to the service and continue running). Then the event is packed into an MPI message and flies away as a request to the services (if the model decided to send the event). Services unpack the event, look at its name, and either process it or ignore.

**Event types**

Now the following types of events are defined (the reaction to them is presented for understanding and is not set in the events themselves, but in their handlers in the classes `Component` and `Service`):

- `READ_FD` – reading from a file (its special cases are `READ_IC`, `READ_CP`).
  Component: send a request to the service, wait for the data, get the data, continue working.
  IOD service: receive a request, take the data from the file, put it into ga-array.

- `SEND_MP` – sending data to mapping and then to another component
  Component: put data into ga, continue working.
  CPL service: take the data from ga, interpolate it to the recipient's grid, put into the recipient's ga.

- `RECV_MP` – receiving mapped data from another component
  Component: wait until the data appears in ga, get it, continue working.
  CPL service: do nothing (everything is already done at the `SEND_MP` step)

- **SAVE_CP** – control point saving
  Component: put data into ga, send a request to the service
  IOD service: receive a request, get data from ga, write to a file.

- **SAVE_DG** – Saving diagnostics (in fact, the same as **SAVE_CP**, but separated for performance reasons)

- **STOP** – normal finish of the model work.
  Component: send a request, continue working
  Services: When the last model sends the **STOP** message, services stop normally.

- **ERROR** – emergency shutdown of the model.
  Component: send a request, stand on hold, because we can not continue working.
  Services: A service that receives this message must shutdown.

**Generators**

In general, since the `Component` class defines the method `raise_event()`, theoretically, event generators are not needed, because at any time in the cycle, you can send a request to the service and it will somehow react. But in practice, this approach means that the user must monitor the time himself and send requests at the right moments. To simplify the life of the user, several event generators are defined in the system, that is, objects that, depending on the model time, issue a request or do nothing.

An example of a generator can be given by the generator of periodic events, e.g., saving diagnosis every 2 days. Such a generator creates an event in 0 hours, 48 hours, 96 hours, ... of model time and does not create anything in the remaining time intervals (for example, at 11 hours 12 minutes of model time)

**How to register an event?**

To register an event it is enough to transfer it to the generator. For this, the user calls the method of the class `Component`, which itself passes it to the right destination (see interface of the `Component`).

**For system developers**

The **VarEvent.f90** file defines the abstract class `VarEvent`, which represents the interface of any generator:

```
! Abstract method that updates the internal state of the generator
! and does (or does not) return an event
procedure(I_update_ve), DEFERRED :: update

! Abstarct destructor
procedure(I_destroy_ve), DEFERRED :: destroy
```

Specific implementations of generators inherit the base class and determine what the object will do when calling `update()`. For example, the `VarEventNormal` class simply checks the proportionality of the current time to the period, which is specified when the generator is initialized.

Next, the `EventScheduler` class creates an array of polymorphic references to all such generators and, whenever the timer proceeds, it queries all the generators if they are ready to issue a request.

**How to define a new event type**

If you want to add a new event, you must:

— In the class `Actions`, add a new event type, its priority, and the corresponding service that will handle it.

— In the class `Event`, add a condition to create your event type. These conditions allow to verify that the event is complete (for example, the mapping event must have a destination, otherwise it can not be processed).

— If you need additional fields that are not in the Event class, add them, making sure that you have implemented their packing to and unpacking from an MPI message.

Now a new type of event is defined in the system. Events with this type can be built and sent. At the same time, in order for the message to actually produce some kind of impact on the system, we need to add event processing to the services and component (see the corresponding sections on services and the component).

**How to make a generator**

To create your generator, you must inherit the base class `VarEvent` and implement the two abstract methods of the base class. In addition, since a pointer is passed to store all the generators, you must provide the pointer to the generator object, for which it's convenient to make a modular function (analogous to `new` in C++). For example, the `VarEventNormal` class is used to generate periodic events:

```
module var_event_normal_module

use var_event_module

type, extends(VarEvent) :: VarEventNormal
private
    integer :: period_sec_ = 0
    type(DateTime) :: start_date_
 contains
    procedure :: update
    procedure :: destroy
end type

 CONTAINS

! Analog of 'new': create a dynamic object and return a reference to it.
! And at the same time we perform the usual functions of the constructor -
! initialize the generator with the start date, event and generation period.
function new_VarEventNormal(ev, start_date, dd, dh, dm, ds) result (obj)
    type(Event), intent(in) :: ev
    type(DateTime), intent(in) :: start_date
    integer, intent(in) :: dd, dh, dm, ds
    class(VarEventNormal), pointer :: obj

    allocate(VarEventNormal::obj)

    obj % ev_ = ev
    obj % start_date_ = start_date
    obj % period_sec_ = dd*60*60*24 + dh*60*60 + dm*60 + ds
end function

! The main function of generation. Depending on the current time
! <cur_time> it generates event <ev>
! and returns .true. or .false.
! In fact, it simply checks the proportionality of the period of generation to the
! difference of current and start time.
logical function update(this, cur_time, ev)
    class(VarEventNormal) :: this
    type(DateTime), intent(in) :: cur_time
    type(Event), intent(inout) :: ev
```

```
    integer(8) :: sec_from_start

    sec_from_start = date2sec(cur_time) - date2sec(this % start_date_)
    update = sec_from_start >= 0 .AND. MOD(sec_from_start, this % period_sec_) == 0

    if (update) then
        ev = this % ev_
    end if
end function

! The object does not contain internal dynamic data, so the destructor is empty.
subroutine destroy(this)
    class(VarEventNormal) :: this
end subroutine

end module
```

Now another type of generator is defined in the system, but the system does not know about it yet. To connect the generator to the system and make life easier for the user, you need to add a simple wrapper to create a new generator in the class `Component`:

```
subroutine register_periodic_event(this, arr_name, act, src, dst, dd, dh, dm, ds)
    class(Component) :: this
    character(*), optional, intent(in) :: src, dst
    character(*), intent(in) :: arr_name, act
    integer, intent(in), optional :: dd, dh, dm, ds
    type(Event) :: ev
    type(ArrayInfo) :: arr_info

    arr_info = this % get_array_info(arr_name)
    ev = Event(arr_info = arr_info, act = act, owner = CompSplitter % i_am(), &
        src = src, dst = dst, file_prefix = this % prefix())

    call this % ev_scheduler_ % add( &
        new_VarEventNormal( ev, ExpInfo % start_date(), &
        MERGE(dd,0,PRESENT(dd)), MERGE(dh,0,PRESENT(dh)), &
        MERGE(dm,0,PRESENT(dm)), MERGE(ds,0,PRESENT(ds))))
end subroutine
```

In the end, the user writes something like:

```
call this % register_periodic_event(arr_name = "test_dg_2D", act = "SAVE_DG", dh = 1)
```

and the wrapper `register_periodic_event()` constructs the event object, defines some default variables, creates the generator object in the dynamic memory, and gives a pointer to it in the object `EventScheduler`.

**Notes for the system developer**

Now for the type of event, you need to know the service that will handle it. This information is not used anywhere, except for the moment when a ga-array is registered in the component, since it must clearly know who to synchronize with. If the issue of explicit synchronization when creating an array in `CommunicatorGA % init_array ()` is resolved, this dependency can be removed altogether.

Perhaps it makes sense to make an analog of JSON for Fortran (FSON).

**5.4 Services**

**General idea**

The Compact Modelling Framework in some form implements a service-oriented architecture (SOA). The idea is that on the side of the client (model) events are generated and corresponding requests (control flow) are sent, to which correspond to the data stored in the Global arrays (data flow). On the server side, some services parse requests from the single queue and perform work (analogous to the pipeline).

Now, the following services are defined:

- `DTR` (distributor) – subscribed to all events, just sends them to all other services. It is necessary for maintaining a single queue in the parallel environment (analogue of the master).

- `IOD` (I/O device) – subscribed to events `READ_FD`, `SAVE_CP`, `SAVE_DG`. When it receives a message, it unpacks it, understands what is required of it (for example, take data from GA named "`test_ga_ocn`" and write to file "`OCN_180x90_tst_DG`") and performs the necessary actions. The other types of messages are simply ignored.

- `CPL` (coupler) – subscribed to event `SEND_MP`. This event determines where to get the data, what to do with it, and where to put it (that is, it's a push event, since events `RECV_MP` are not required to process it).

**How it works**

To simplify the creation of a new service, a basic abstract class `Service` is implemented, which has the following interface:

```
! Base class constructor
procedure, public :: init_base

! Base class destructor
procedure, public :: destroy_base

! Main cycle of events processing
procedure, public :: request_cycle

! Virtual method for processing one event
procedure(I_handle_request), private, DEFERRED :: handle_request

! Virtual constructor
procedure(I_init_service), public, DEFERRED :: init

! Virtual destructor
procedure(I_destroy_service), public, DEFERRED :: destroy
```

The main programm **cpl_main.f90** contains the following lines:

```
select case(CompSplitter % i_am())
    case("DTR")
        allocate(ServiceDTR :: service_p)
    case("CPL")
        allocate(ServiceCPL :: service_p)
    case("IOD")
        allocate(ServiceIOD :: service_p)
end select

! Start model cycle
```

```
if (CompSplitter % is_model()) then
    call comp_p % model_cycle()
else
    call service_p % init_base(comm)
    call service_p % init()
    call service_p % request_cycle()
end if
```

That is, every process that belongs to the group of processes of a certain service (defined in `CompSplitter`), allocates its polymorphic pointer and then calls the constructor and enters the event processing cycle `request_cycle()`. (At the end of the program, service destructors are called in the same way)

The base class method `request_cycle()` contains two identical loops, one for registering arrays, and the second for real event processing. The structure of the loop is simple: for the time being there are working models, accept the request, call the virtual method `handle_request(ev)` and track the `STOP` signals from the models.

```
do while (this % running_count_ > 0)

    ! Receive any request
    call this % receive_request(ev)
    call this % handle_request(ev)

    select case(ev % action())
        ! One component finish work
        case("STOP")
            this % running_count_ = this % running_count_ - 1
            CYCLE
    end select
end do
```

**How to make a new service**

- Create a skeleton of the derived class

As a result, to create a service, you need to inherit the class `Service` and define three virtual methods: `init`, `destroy`, `handle_request`.

```
module service_tst_module

use utils_module
use actions_module
use service_module
use event_module
use communicator_ga_module
use component_splitter_module

implicit none

type, extends(Service) :: ServiceTST
private

  contains
    !=====================
    ! ===== PUBLIC API =====
    procedure, public :: init => init_tst
```

```
    procedure, public :: destroy => destroy_tst
    procedure, public :: handle_request => handle_tst

    procedure, private :: handle_my_method1
    procedure, private :: handle_my_method2

    ! ===== PUBLIC API =====
    !=====================
end type

 CONTAINS

subroutine init_tst(this)
    class(ServiceTST) :: this
    ! Your constructor
end subroutine

subroutine destroy_tst(this)
    class(ServiceTST) :: this
    ! Your destructor
end subroutine

subroutine handle_tst(this, ev)
    class(ServiceTST) :: this
    ! Code of event handler ev (read further)
end subroutine

! Rest methods
end module
```

- Fill in the class

We fill the `init_tst`, `destroy_tst` methods with the necessary actions. Next, fill in the main method – `handle_tst`. Under the current agreement, when the service "sees" the array for the first time, the method must register this communication channel (that is, the ga-array) in the communicator. For this you can use the following construction:

```
if (.NOT. this % comm_ % is_registered(ev % ga_name())) then
            call this % try_register_service_ga(ev)
...
! Other actions required for the first time when you receive a message of this type
! ( i.e. by this ga-channel)
end if
```

If the array is already registered, you can immediately deal with its processing. As an example, the `handle_request` method of the class `ServceIOD` is shown below. It handles only events related to working with files and the error message `ERROR` (which simply leads to an abnormal termination). Other events are ignored. During the first reception, the array is registered, during the rest it is processing the event and outputting information to stdout with the built-in auxiliary method of the base class `report_handle()`. The methods `handle_put()`, `handle_get()` contain the real logic of extracting an array from ga and writing it to a file using `FileHandler_NC`.

```
subroutine handle_iod(this, ev)
    class(ServiceIOD) :: this
    type(Event), intent(in) :: ev
```

```
      select case (ev % action())
          case("SAVE_CP", "SAVE_DG")
              if (.NOT. this % comm_ % is_registered(ev % ga_name())) then
                  call this % try_register_service_ga(ev)
              else
                  call this % handle_put(ev)
                  call this % report_handle(ev)
              end if

          case("READ_FD")
              if (.NOT. this % comm_ % is_registered(ev % ga_name())) then
                  call this % try_register_service_ga(ev)
              else
                  call this % handle_get(ev)
                  call this % report_handle(ev)
              end if

          case("ERROR")
              call this % report_handle(ev)
              call exit(1)
      end select
end subroutine
```

Warning: mind the synchronization issues! The service should be blocked if the ga-array (which is the exchange buffer) is still occupied (for the "put data" event) and, conversely, is free (for the "get data" event). For this, the status of the information array is checked. If you do not set the lock, there is no guarantee that the resulting data will be complete. After the service has put or get the data, it must change the status of the ga-array accordingly.

For example, when `ServiceIOD` receives the request `SAVE_DG`, it knows (see the corresponding handler in `Component`) that the data is already completely in the ga-array, so no additional checks are needed. When the service is ready to release the ga-array (in the method `handle_put()`) after the data has been copied into its memory, it synchronizes and marks the array as free:

```
call this % comm_ % sync(service_id)
if(CompSplitter % is_first_rank()) call this % comm_ % put_info(ev % ga_name(),&
    COMM_GA_STATUS, "free")
```

- Register the service in the system

Warning: In this paragraph, changes are made to the system code. If you are not sure about your actions, contact the developer.
At the moment the system does not know anything about the new service (call it `TST`), so you need to:
1) In the file `ComponentSplitter` add the necessary service names to the arrays of components and services:

```
character(3), parameter :: COMPONENT_NAMES(9) = &
    (/ "DTR", "CPL", "IOD", "OCN", "ATM", "ICE", "LND", "SEA", "TST"/)
character(3), parameter :: SERVICE_COMPS(4) = &
    (/ "DTR", "CPL", "IOD", "TST"/)
```

The class performs division into groups of processes in the multiprocessor environment, and now every process can find out if it belongs to the group, for example, of the `CPL`.
2) Connect the service module in the `cpl_main` and define the creation of a real service object using the previous item:

```
select case(CompSplitter % i_am())
...
    case("TST")
        allocate(ServiceTST :: service_p)
...
end select
```

Now, if you specify `TST 2` at startup, the system will start the new service on 2 processes.

- Send right requests from the client

Now the service is fully operational – it starts and accepts requests from the client (for the time being it's just notification about the end of the run `STOP`). In order for the client to generate the right events, it is necessary to define a new event, to make a generator for it, and to describe the actions necessary from the client. How to do this is described in the Model Component section.

**Notes for the system developer**

If in the future there will be an opportunity to get rid of explicit synchronization through an array of information, it will be good. Now this approach is chosen, since we can not "lose data". That is, even if some component (ocean model) is faster than another component (atmosphere model or IOD, which slowly writes data to a file), we do not have the right to lose the array. The accumulation of arrays in the form of a queue will also lead to nothing, since models usually work at a constant speed and as a result, the queue will simply exhaust all available memory. Therefore now, if the "fast" model is ready to put data, but the ga-buffer is still occupied, it is blocked.

**5.5   Working with NetCDF-files**

**General idea**

NetCDF is a hardware-independent self-describing format and a set of libraries for working with it. NetCDF is the actual standard for storing geophysical data. As a result, to save, for example, an array of speeds, you do not need to invent your procedures with a heap of read/writes, but just call the ready function of the NetCDF library. An important property of the procedures is that they can be performed in both sequential and parallel modes.

NetCDF has a rather high-level interface in terms of operations on files, but rather low-level from the user's point of view, since it is necessary to understand the intricacies of certain built-in procedures. In this case, the control over the correctness of all operations lies entirely with the user. Since it is often necessary to work with NetCDF files, there is a desire to create a helper class that will have a high-level interface, hiding all the complexities of NetCDF within itself. So the class `FileHandler_NC` appeared. Firstly, it simplifies the work with NetCDF, and secondly, it adds some functionality. For example, the class provides a convenient way to access data not only by index, but also by timestamp and an ability to read the time axis of files in different formats. To work with the class, it's enough to link in the module `file_handler_nc_module`, create an instance of the class, and use it to manage the file.

`FileHandler_NC` is used in:

— `Service_IOD` for parallel put/get-operations

— `Component` for analysis of a file with time axis

— `Offline` to create initial condition files

— `Service_CPL` to read interpolation weight files

— in data assimilation system, etc.

As a result, all operations with NetCDF of the whole system are delegated to the helper class, which greatly simplifies the code by encapsulating all the logic in one place.

**Work example**

Different ways of working with the class can be found in the test **Coupler/test/test_filehandler_nc.f90**. For example, the standard scheme of work is: create a handler, use it to create a file and variables, write data.

```
use file_handler_nc_module

type(FileHandler_NC) :: handler

call handler % create_file("test_2D_dt.nc", mpi_comm = tm % comm())
call handler % create_dim("i", il)
call handler % create_dim("j", jl)
call handler % create_dim("k", kl)
call handler % create_time_dim()

call handler % create_var("test_2D_dt", "real4", "i", "j", dimt_name = "TIME")

call handler % put(arr_2D, lo = decomp % lower_bound_2D(), &
    hi = decomp % upper_bound_2D(), dt = DateTime(1988, 03, 15, 0, 0, 0))

call handler % close_file()
```

**Description of API**

**Create/open/close file**

```
create_file(filename, mpi_comm)

Description:       Create file or rewrite previous
Parameters:
  filename        - string name of file to create
  mpi_comm        - <optional> MPI mpi_comm if it is parallel run
```

```
open_file(filename, mpi_comm, status)

Description:       Try to open file
Parameters:
  filename        - string name of file to open
  mpi_comm        - <optional> MPI mpi_comm if it is parallel run
  status          - <optional> status of operation: 0 if ok, 1 is error
```

```
open_or_create_file(filename, mpi_comm)

Description:       Try open and then create file
Parameters:       Combination of parameters for open_file and create_file procedures
```

```
close_file()
Description:       Close current file
Parameters:
```

**Create dimensions**

```
create_dim(name, length)
```

```
Description:      add NC-dimension to file
Parameters:
  name            - name of dimension
  length          - corresponding length of dimension
```

```
create_time_dim()

Description:      add time NC-dimension to file
Parameters:
```

**Create/open variables on dimensions**

```
create_var(var_name, var_type, dim1_name, dim2_name, dim3_name, dimt_name)

Description:      Try to create var. Error if this var is already exist.
Parameters:
  var_name        - string name of file to create
  var_type        - type of variable
  dim*_name       - create variable of these dimensions
```

```
open_var(var_name, status)

Description:      Try to open variable
Parameters:
  var_name        - string name of variable to open
  status          - <optional> status of operation: 0 if ok, 1 is error
```

```
open_or_create_var(this, varname, var_type, dim1_name, dim2_name,
                   dim3_name, dimt_name)

Description:      try open and then create variable
Parameters:      combination of parameters for open_var and create_var procedures
```

**Write/read variables**

```
put/get (arr, lo, hi, dt)

Description:       put and get data.
Parameters:
  arr    - data array of supported type and dimension
           (int4, int8, real4, real8, 1D, 2D, 3D)
  lo, hi - lower and upper bounds of dimensions (e.g. (/1, 1/), (/ il, jl /) )
  dt     - <optional> DateTime corresponding to field. Necessary for time vars.
```

**Various operations**

```
put_att/get_att(att_name, att_val, is_global)

Description:       put/get attribute to variable or whole file
Parameters:
  att_name        - name of attribute
  att_val         - value of attribute of supported type (int, character)
  is_global       - <optional> if .TRUE., this attribute is made NF90_GLOBAL
```

```
function get_dim_size(dim_name)
Description:       Return size of interested dimension
Parameters:
  dim_name         - dimension name
```

```
get_time_axis(time_axis)
Description:       Get time axis
Parameters:
  time_axis        - integer(8), allocatable :: time_axis(:) - where to put time axis
                     in seconds from DateTime % epoch_start()
```

```
logical is_time_var()
Description:       Check if current variable has a time axis
Parameters:
```

**5.6  GA-communicator**

The general idea is to allow the user to easily access different parts of a distributed array. This is done using the abstraction of PGAS (Partitioned Global Adress Space). PGAS suggests that there is some virtual huge array that is accessible from any process that participated in its creation. Of course, in fact, there is no global array, and its parts are stored in processes' memory, but the user does not know about it – all the subtleties are taken over by the library, which is why simplicity is achieved. For example, a client at process 12 can ask for an item with indexes [124, 97], as if it has direct access to it. Behind the scenes, PGAS will know which process the item belongs to (for example, the 18th), execute the MPI request for it, get the result and return it to the client.

An implementation of PGAS abstraction is the Global Arrays (GA) library (which, by the way, is also installed by the geophysical software installation script). There are also other implementations. Finally, the `Communicator_GA` class is a class of the CMF system, representing a kind of facade for this library, that is, it defines an even higher-level interface and hides some of the subtleties of the GA.

**Interface**

```
subroutine init(max_index, proc_local_count)

Description:               construct communicator object for <num_of_sides> components
Parameters:
    max_index             - maximum index of component, which will be used for work
                            with object (normally equal to number of defined comps)
    proc_local_count      - size of local communicator, required for agile memory
                            allocation
```

```
subroutine init_group(src_id, src_ranks, dst_id, dst_ranks)

Description:               register processor group between two sides
Parameters:
    src_id, dst_id        - ids of sides
    src_ranks, dst_ranks  - ranks of all processes of sides
```

```
subroutine init_array(arr_name, datatype, dimnum, holder_id, holder_decomp, &
                       subscriber_id)
```

```
Description:            initialize array based on Decomposition object, it can be
                        accessed from <holder_id> and  components,
                        but stored on <holder_id>. If array with such name already
                        exists - delete previous and create new.
Parameters:
    arr_name            - string name of array
    datatype            - supported datatype string: "real4", real8", "int4"
    dimnum              - supported dimension string: "2D", "3D"
    holder_id           - id of source component who hold array in memory
    holder_decomp       - decomposition of holder side
    subscriber_id       - id of subscriber component
```

```
subroutine init_array(arr_name, datatype, dim1_len, dim2_len, dim3_len, holder_id, &
                  holder_size, subscriber_id)

Description:            initialize array based on dimension sizes, it can be
                        accessed from <holder_id> and  components,
                        but stored on <holder_id>. If array with such name already
                        exists - delete previous and create new.
Parameters:
    arr_name            - string name of array
    datatype            - supported datatype string: "real4", real8", "int4"
    dimnum              - supported dimension string: "2D", "3D"
    dim*_len,           - size of each dimension
    holder_id           - id of source component who hold array in memory
    holder_size         - how many processors owns the GA
    subscriber_id       - id of subscriber component
```

```
subroutine destroy_array(arr_name)

Description:      destroy global array (you should set appropriate group before this
                 call - the same as on init_array)
Parameters:
    arr_name          - string name of array
```

```
subroutine sync(src_id, dst_id)
Description:            barrier for src_id, dst_id
Parameters:
    src_id, dst_id       - indexes of groups
```

```
integer function id(arr_name)

Description:            return ga_id of array with given name.
                       Return -1 if no such array.
Parameters:
    arr_name            - string name of array
```

```
subroutine put (arr_name, lo, hi, arr)
subroutine get (arr_name, lo, hi, arr)

Description:            put/get data
Parameters:
    arr_name            - string name of array
```

```
     lo, hi                 - arrays representing area (in global indexing) you
                              want to put/get
     arr                    - your buffer for data
```

**Examples of usage**

In more detail, examples of use can be found in tests for the class
(**coupler/test/ga_communicator**). Below are the popular examples taken just from there.

**Creating an array shared by two components**

The class allows to create an array that will be distributed on one component, but still visible to another component. This allows, for example, to create a temperature array that will be physically distributed over the ocean's cores (and they can put and get data from it), but, in addition, the service of the coupler can also work with this array, although it does not store any part of it.

```
! Ask the CompSplitter for component identificators
ocn_id = CompSplitter % comp_id("OCN")
cpl_id = CompSplitter % comp_id("CPL")

! Initialize the communicator object with the number of components and the local
! communicator size of each component
call comm_ga % init(CompSplitter % comp_defined(), CompSplitter % comm_local_size())

! Ask the CompSplitter for process lists of each components
call CompSplitter % proc_list("OCN", list = proc_list_ocn)
call CompSplitter % proc_list("CPL", list = proc_list_cpl)

! With their help, register the group [ocn_id, cpl_id]
call comm_ga % init_group(ocn_id, proc_list_ocn, cpl_id, proc_list_cpl)

! The ocean creates its decomposition and distributes it to all -
! it is necessary that all interested core groups call registration
! of the array with the same parameters.
! In a real program, you can take decomposition data from the global
! ModelInfo array that stores information about all components
if (CompSplitter % i_am() == "OCN") then
    ocn_decomp = Decomposition(il, jl, kl, "2D", CompSplitter % comm_local_size(), &
        CompSplitter % rank_local())
end if

call ocn_decomp % broadcast(CompSplitter % proc_first("OCN"), &
    CompSplitter % comm_world())

! Finally, we register the array, indicating the ocean as the holder,
! passing its decomposition (so that the GA allocates the array exactly so),
! and the subscriber is the coupler
call comm_ga % init_array(arr_name = "glob_2D", datatype = "real4", dimnum = "2D", &
    holder_id = ocn_id, holder_decomp = ocn_decomp, subscriber_id = cpl_id)

! Now you can put data into the array: for example, let only the ocean do it,
! so that each core puts a global array (nonsense in a real program)
if (CompSplitter % i_am() == "CPL") call comm_ga % put("glob_2D", (/ 1, 1 /), &
    (/ il, jl /), tgd % glob(:,:,1,1))
```

```
! We necessarily perform synchronization, that is, we wait till all the
! data has been put, since the put/get calls are nonblocking
call comm_ga % sync(ocn_id, cpl_id)

! Now we can take the data: all the cores of both components take local pieces
! and compare them with the predefined test
call comm_ga % get("glob_2D", (/ w, s /), (/ e, n /), tgd % loc_2D)
call tm % assert(tgd % is_correct(arr_type = "2D"), "all get 2D local patch &
    defined in parameters")
```

**Creating an array that is shared by only one component**

Sometimes you need to create an array for use only within one component. In this example, we will create such an array for the ocean component. In addition, instead of passing the ocean decomposition to the procedure, we simply give the dimensions so that the class itself decomposes the array for us. Most steps repeat the previous example, except that the group and the array are now created for the same identifiers (the holder and the subscriber are the same), and the version of the procedure for registering the array without indicating the decomposition is called.

```
! Ask the CompSplitter for component identificator
ocn_id = CompSplitter % comp_id("OCN")

! Initialize the communicator object with the number of components and the local
! communicator size of each component
call comm_ga % init(CompSplitter % comp_defined(), CompSplitter % comm_local_size())

! Ask the CompSplitter for process list of the component
call CompSplitter % proc_list("OCN", list = proc_list_ocn)

! Register the group for ocean only
call comm_ga % init_group(ocn_id, proc_list_ocn)

! We register the array without specifying a subscriber -- this will be the
! component itself. In addition, we only transfer the dimensions of the array il,jl
call comm_ga % init_array(arr_name = "priv_2D", datatype = "real4", dim1_len = il, &
    dim2_len = jl, holder_id = ocn_id, &
    holder_size = CompSplitter % comm_local_size("OCN"))

! Put data (again global)
call comm_ga % put("priv_2D", (/ 1, 1 /), (/ il, jl /), tgd % glob(:,:,1,1))

! Synchronize to make sure that everyone put the data
call comm_ga % sync(ocn_id)

! Get the local data
call comm_ga % get("priv_2D", (/ w, s /), (/ e, n /), tgd % loc_2D)
```

**5.7   Additional tools for the model**

The tools listed in this section are not the logical part of the CMF, but represent some external tools that the client code can use. For example, although halo exchanges or reduce-operations are not necessary for all models, CMF contains them as separate "convenient" functions that work on the communicator of the calling model and are not visible to the rest of the system.

**Halo updater**

**General idea**

In many models, there is a need to exchange the border cells of the local calculation domain of each process with the neighboring processes. This task is solved by the class `HaloUpdater`. Formally, it is not a part of the CMF, but is a separate module that any model can use. Now the exchange functions are implemented for latitude-longitude and bipolar grids (both for T- and V-cells), for 2D-, 3D-arrays of any kind used in the ocean model. Structurally, the module consists of a template class `HaloUpdaterBase` (without specifying the types), which does the basic work, and the `HaloUpdater` class, which presents the high-level interface to the client and calls specific methods of the low-level class.

From the user's point of view, when you connect the `halo_updater_module`, the `HaloUpdateMaster` object becomes available, which is used for exchanges.

**Example of usage**

Details of operation of the updater can be found in the test (**coupler/test/halo_update**). Below is a typical example of use. Note the optional parameter
`change_size_on_bipolar`: if it is equal to the .true., the sign will be changed. In the opposite case (or if it is not specified), the sign will be saved. In addition, for convenience, all procedures have the same form for all types of arrays (this is called function overloading).

```
use halo_updater_module

! Create a decomposition object
ocn_decomp = Decomposition(180, 90, 20, "2D", comm_local_size, rank_local, &
    is_icycle = .true., is_bsc = .true.)

! Initialize the master, specifying the model decomposition and
! the maximum halo width
call HaloUpdateMaster % init(decomp = ocn_decomp, max_halo_width = 2)

! Ready to exchange: in this case we exchange a three-dimensional array
! of temperatures with halo width 1
call HaloUpdateMaster % update(t_c(:,:,:,1), update_width = 1, grid_type = 'T')

! And now a two-dimensional array u_c(1,:,:) with halo width 2
call HaloUpdateMaster % update(u_c(1,:,:), update_width = 2, grid_type = 'V', &
    change_sign_on_bipolar = .TRUE.)
```

**Other utilities**

**UtilsAllReduce**

Returns the global sum of a variable on the communicator or, in the presence of weights, a weighted sum. Warning: Be careful with global operations – they can lead to performance degradation.

```
use utils_module

real(kind=8) :: my_global_sum, my_global_int

! Calculate the sum over the ocean communicator for the variable some_local_val
my_global_sum = UtilsAllReduce(local_val = some_local_val, comm = ocn_comm)

! Calculate the area-weighted sum over the ocean communicator for the variable
! some_local_int
my_global_int = UtilsAllReduce(local_area = some_area, local_val = some_local_int, &
```

```
      comm = ocn_comm)
```

**PointSaver**

Saves a data point along with the corresponding timestamp.

```
use point_saver_module

type(PointSaver) :: ps
real(8) :: some_val

! Initialize the object with the name of the file (and the same name as the variable
! inside it) and the period of data flush to disk.
call ps % init(varname = "var1", flush_period = 100)

! Put the data to the file together with the current time
! (in this case it is requested from the component via the method model_time())
call ps % put(some_val, cmp_ptr % model_time())

! Do not forget to call the destructor at the end (close the file)
call ps % destroy()
```

**6   Short instructions**

Here are brief lists, which you should not forget about while reconfiguring the model.

**6.1   Configuration switch**

— Number of ice cores is specified in **ice_in** and **comp_ice**

— Ice timestep is specified in **ice_in** and **ice_list.in**

— Forcing data bases are chosen in **atm_list.in** and **lnd_list.in**

— The formula of freezing point in **ice_module.f90** is desirable to be the same as in the used thermo-dynamics scheme of CICE (by default – mushy).

— After each reconfiguring, affecting the CICE settings, completely rebuild the coupled model.

— If the reduced ice grid is used then in **ice_list.in** and **config** ther must be specified the reduced j-size of the domain.

— Ocean-ice coupling frequency is specified in two places: in **o_tf_module.f90** and **ice_cice_driver_module.f90**.

**6.2   Deep reconfiguring of the forcing under CMF2.0**

— Forcing in all components (atmosphere, land, ...) should start with the same date.

— The start date in **run_list.in** must coincide with the start date of the forcing database.

— Check the tuning of `time_start_min`

— Check the in situ – potential temperature conversion

— Check the sea level initialization

— Check whether the SST relaxation is on or off

— When registering reads from files (`ACTION_READ_FD`) within each component, all reading periods must be a multiple of the minimum of the reading periods of this component. In particular, because of this, either the periods of reading the average monthly values (runoff, rain, etc.) have to be 30.5 or even 30 days instead of 1440 * 365/12 minutes, or you need to enter an additional probe value, read with a period of 2 hours .

**7 Elements of numerical and program implementation**

**7.1 Notes on the differences between CMF2.0 and CMF3.0**

In both versions of the system, the step counters `time_l` and `time_l_in_run` are initialized by value of 1 at the start from the initial conditions file and incremented outside the `o_ driver_module`. They represent the step number that is being taken now (that is, for the ocean, they mean how many steps will be taken when the current step of the physical model is completed). The difference is that for CMF2.0 the whole calendar is calculated basing on these counters and, accordingly, `time_min`, `time_hour`, ... is the moment of the end of the current model step. In CMF3.0, these variables (`time_min`, `time_hour`, ...) are not directly calculated by the coupler and made as an additive, for compatibility with CMF2.0. And they do not correspond to the end, but to the beginning of the current step.

**A Appendix: namelist parameters**

Table 1: Namelist parameters

| File | Varable | Possible values | Comments |
|------|---------|-----------------|----------|
| atm_list.in | atm_forcing_type | 1 | "normal" year cycle CNYFv2 (CORE-I) |
| | | 2 | "real" IAFv2 data for 1948-2009 (CORE-II) |
| lnd_list.in | atm_rivers_type | 1 | "normal" year cycle CNYFv2 (CORE-I) |
| | | 2 | "real" IAFv2 data for 1948-2009 (CORE-II) |

**B Appendix: standard configurations**

**laptev0125c**

Test configuration for PC. Computational domain of the size **40 × 60** in the region of the Laptev Sea with the added artificial ring island. The grid is latitude-longitudinal, the resolution is about **0.125°**. The ice grid is full. Forcing CNYFv2. The turbulence coefficients are small, close to the global eddy-resolving settings.

**arctic025t**

Arctic starting from **50°** N with **0.25°** resolution. Computational domain of **1440 × 160**, three-polar grid. The ice grid is full. Forcing CNYFv2. Viscosity is only biharmonic, diffusion is 300, of the NEMO type. Implicit Coriolis approximation.

---

## Author Response (AR2)

Authors' reply to the 2nd review of "Compact Modeling Framework v3.0 for high-resolution global ocean-ice-atmosphere models" by Kalmykov et al

Maxim Kaurkin

**maksim.kaurkin@phystech.edu**

We thank the reviewers for the second reading of the article and further suggestions for its improvement. We have made the proposed corrections. Some details are given below in the point-by-point response to the reviewers' comments (shown in "italic").

**Referee comment**

*• P.3 L.29 : I still think that it is not appropriate and confusing to write that OASIS3-MCT coupling library has a non-coupler design. You could simply state "Unquestionable advantage of the OASIS (OASIS3, OASIS4 and OASIS3-MCT) design …" ; or at least use "… no-standalone-coupler design …"*

**Author's response**

We have used the second variant.

**Referee comment**

*• P.3, L.30-33: OASIS3-MCT primary function is coupling not I/O; it is a bit unfair to turn down OASIS3-MCT because of its non-optimal I/O system. Sentences like "But still the system contains master-process I/O routines, which obviously limit its use for large grids" and "Nevertheless, such behaviour could be fully acceptable for runs with non-intensive mapping and I/O" are confusing because of course OASIS3-MCT use is not limited for coupling or mapping of large grids. Can you rephrase these sentences taking this remark into account?*

**Author's response**

We have rephrased these sentences as "But still the system contains master-process I/O routines, which can be a limiting factor in case of intensive data dumping" and "Nevertheless, such behaviour does not limit the coupling and mapping on large grids".

**Referee comment**

*• P.4, L.1: In your reply, you write that you have changed the "According to proposals of the Earth System Modeling conference …", but you have not. The mentioned workshop was not named like that. Please change for "According to the analysis of the coupling technologies for Earth System Modeling workshop (Valcke et al., 2012)…"*

**Author's response**

We have applied this correction.

**Referee comment**

*• P.4, L.13 and in the abstract: what is precisely the meaning of "abstract" in "high-level abstract driver"? I would say that if a high-level driver in implemented in CMF2.0 (similar to the CESM driver) it is not abstract at all. Please modify or clarify.*

**Author's response**

We have explained this term in the last sentence of the Section 3.1: "Also the addition or modification of components does not affect the main CMF code, because it implements the abstract driver and the abstract component, which do not mention any specific component names (ocean, atmosphere, ice, etc)."

*Referee comments*

*• P.6, L.8: Please add "During the run" before "The data intended for mapping is sent …"*
*• P.6, L. 33-34: change "which corresponds to 240 × 9 atmosphere-ocean mappings and 120 × 3 ocean-atmosphere mappings" for "which corresponds to 120 × 3 ocean-atmosphere mappings and 240 × 9 atmosphere-ocean mappings" as this is the order used on L.29.*
*• P.7, L.16: Please make a new paragraph before "Results …"*

**Author's response**

We have applied these corrections.

*Referee comment*

*• P.8, L. 5-6: You write "For eight coupler cores this difference becomes 13% and 22%". How can this be as for 8 CPL cores, the number from Table 1 is 116 for both (8640,3456) and (10368,4320). I think something is wrong with the numbers reported in Table 1 as they are the same for 4 and 8 CPL cores for both (8640,3456) and (10368,4320), while this does not seem to be the case on Figure 4. Finally, the paragraph would be easier to understand if Table 1 would provide also numbers for (1152, 288).*

**Author's response**

This was a mistake, thank you! The table has been corrected and extended with results for the (1152, 288) decomposition.

*Referee comment*
*• P.17, L.19: "flattering" should be "some flattening"*

*(also mentioned by* Referee #1)

**Author's response**

Corrected.

*Referee comment*
*• P.19, L.15: can you give more details on "but greatly simplifies code"; which code?*

*The CMF3.0 code? The coding one has to do in his component to interface with CMF3.0?*

**Author's response**

It is the CMF3.0 code. We have moved this issue to the previous paragraph, rephrasing it as "...which allows one to divide the coupler responsibilities into small separate services, easily plug/unplug them and add new ones to the system..."

*Referee comments*
*The following typos and English formulations should be corrected:*
*• P.6, L.7: "by by means"*
*• P.6, L.8: "In the beginning …" should be "At the beginning …"*
*• P.6, L.15: "The process is performed in the SCRIP format, i.e. …" could be "The SCRIP format is used, i.e. …"*
*• P.6, L.27: "maintaining" should be "ensuring"*
*• P. 10, L.14: consider changing "… to accelerate until the writing time is less then …" for "… to accelerate as long as the writing time is less than or equal to …"*
*• P.10, L.14: consider changing "… so it allows to obtain some acceleration of writing" for "so a reduction in the writing time is obtained when increasing the number of cores"*

**Author's response**

These issues have been corrected according to the reviewer's suggestions

*Referee comment*
*• P.10, L. consider changing "is more of a nice result" for "is a better result"*

**Author's response**

Changed to "is rather a nice particular result than a permanent CMF feature".

*Referee comment*
*• P.14, L.30: "are weaker" could be "are not as good as"*

**Author's response**

This correction has been applied

*Referee comment*

*• P.15, L.5": consider changing "In the CMF3.0, services are responding to messages during all run time. Events are requests for certain actions on data arrays." for "In CMF3.0, services respond to request messages and trigger certain actions on data arrays."*

**Author's response**

Changed to "At any moment during the run time, the CMF3.0 services can respond to request messages and trigger certain actions on data arrays. So, the model is allowed to send such requests (i.e. raise events) unexpectedly...".

*Referee comments*

• *P.15, L.6-7: "we may know a schedule" could be "we may know in advance a schedule"*
• *P.15, L.18: consider changing "ga-array" for "GA-array*
• *P.17, L.11: "turns off" should be "turned off"*
• *P.18, L.7: consider changing "The restriction of spatio-temporal resolution was implied by available computer resources and not by restrictions of the CMF." for "The spatio-temporal resolution was restricted by available computer resources but not by CMF itself."*
• *P.18, L15: please add commas before and after "along with intraannual distribution characteristics of monthly data fields"*
• *P.18, L.28-29: consider changing "the solution of the data assimilation problem is scaled almost linearly" for "the data assimilation problem scales almost linearly"*
• *P.19, L.4: add "the" before "coupler"*
• *P.19, L.12: consider removing "strong" in "nearly linear strong scalability"*
• *P.19, L. 19: change "functional" for "functionality"*

**Author's response**

These issues have been corrected according to the reviewer's suggestions

[revised manuscript text omitted]

---

## Author Response (AR3)

Author's Response to reviewers' comments on "Compact Modeling Framework v3.0 for high-resolution global ocean-ice-atmosphere models" by  Vladimir V. Kalmykov et al.

Maxim Kaurkin

**maksim.kaurkin@phystech.edu**

Dear Topical Editor and Editorial Support, thank you very much for our manuscript acceptance for final publication in GMD. Two misprints were fixed after our last submission (18 Aug 2018, see latexdiff below).
Please note that our affiliation had been changed (new organization had been added, see latexdiff below) as well as "Acknowledgments" section due to the fact that new researches were carried out for the publication after its initial submission (17 Nov 2017) .

Regards,
Authors
Vladimir V. Kalmykov
Rashit A Ibrayev
Maxim N.Kaurkin
Konstantin V. Ushakov

**Compact Modeling Framework v3.0 for high-resolution global ocean-ice-atmosphere models**

Vladimir V. Kalmykov[1,3], Rashit A. Ibrayev[1,2,3,4,5], Maxim N. Kaurkin[1,3,5], and Konstantin V. Ushakov[1,2,3,5]

[1]  Hydrometcenter of Russia, B. Predtechensky per., 11-13, Moscow, 123242, Russia
[2] Marchuk Institute of Numerical Mathematics, Russian Academy of Sciences, ul. Gubkina, 8, Moscow, 119333, Russia
[3] Shirshov Institute of Oceanology, Russian Academy of Sciences, Nahimovskiy prospekt, 36, Moscow, 117997, Russia
[4] Moscow Institute of Physics and Technology (State University), Institutskiy per. 9, Dolgoprudny, Moscow oblast, 141700, Russia
[5] Federal State Budget Scientific Institution "Marine Hydrophysical Institute of RAS", Kapitanskaya str., 2, Sevastopol, 299011, Russia

**Correspondence:** Maxim N. Kaurkin (maksim.kaurkin@phystech.edu)

*Acknowledgements.* The research of Sections 1–3, 5.1 and 5.2 was  supported by the Russian Science Foundation (project no. 14-37-00053) and performed at the Hydrometeorological Research  Center of the Russian Federation. The research of Sections 4 and 5.3 was supported by the Russian Science Foundation (project no. 17-77-30001) and performed at the Federal State Budget Scientific Institution "Marine Hydrophysical Institute of RAS".